# Unsupervised Federated Graph Learning

**Lele Fu**[1], **Tianchi Liao**[1], **Sheng Huang**[1], **Bowen Deng**[1], **Chuanfu Zhang**[1],
**Shirui Pan**[2], **Chuan Chen**[1]*

[1]Sun Yat-sen University, Guangzhou, China
[2]Griffith University, Brisbane, Australia

`{fulle,liaotch,huangsh253,dengbw3}@mail2.sysu.edu.cn`
`s.pan@griffith.edu.au, {zhangchf9,chenchuan}@mail.sysu.edu.cn`

## Abstract

Federated graph learning (FGL) is a privacy-preserving paradigm for modeling distributed graph data, designed to train a powerful global graph neural network. Existing FGL methods predominantly rely on label information during training, effective FGL in an unsupervised setting remains largely unexplored territory. In this paper, we address two key challenges in unsupervised FGL: 1) Local models tend to converge in divergent directions due to the lack of shared semantic information across clients. Then, how to align representation spaces among multiple clients is the first challenge. 2) Conventional federated weighted aggregation easily results in degrading the performance of the global model, then which raises another challenge, namely how to adaptively learn the global model parameters. In response to the two questions, we propose a tailored framework named FedPAM, which is composed of two modules: Representation Space Alignment (RSA) and Adaptive Global Parameter Learning (AGPL). RSA leverages a set of learnable anchors to define the global representation space, then local subgraphs are aligned with them through the fused Gromov-Wasserstein optimal transport, achieving the representation space alignment across clients. AGPL stacks local model parameters into third-order tensors, and adaptively integrates the global model parameters in a low-rank tensor space, which facilitates to fuse the high-order knowledge among clients. Extensive experiments on eight graph datasets are conducted, the results demonstrate that the proposed FedPAM is superior over classical and SOTA compared methods.

## 1 Introduction

In response to growing concerns over privacy protection [1, 2, 3], the storage of graph data is becoming increasingly decentralized, where individual clients hold their own private subgraphs. Nevertheless, distributed subgraphs result in the emergence of data silos, triggering data unavailability and impeding the generalization of graph neural networks (GNNs) [4, 5, 6, 7, 8]. In this context, federated graph learning (FGL) [9, 10, 11, 12, 13, 14] has emerged as a paradigm that enables the collaborative training of a powerful GNN across multiple clients, while maintaining the usability of private subgraphs without compromising their confidentiality.

FGL has made substantial progress in recent years. In the context of graph-level tasks, studies [15, 16, 17] primarily concentrate on handling the challenges arising from distributional heterogeneity. For node-level tasks, considerable attentions [18, 9, 19] are devoted to defining the topological heterogeneity in distributed subgraphs and overcoming its adverse effects on the effectiveness of federated training. The existing approaches provide innovative insights and solid solutions for FGL.

---

*Corresponding author.

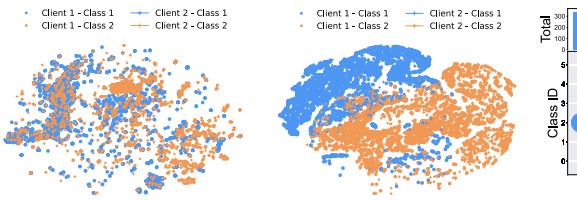

(a) FedAvg       (b) FedPAM

Figure 1: Scatter comparison on Physics dataset for FedAvg and FedPAM. It can be seen that FedPAM achieves a better inter-class separation, aligning the representation spaces of different clients, while FedAvg fails to do so.

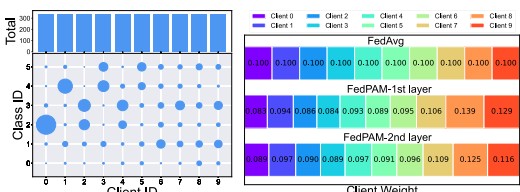

(a) Statistics of CiteSeer    (b) Comparison of client weights

Figure 2: Comparison of client weights on CiteSeer dataset for FedAvg and FedPAM. FedPAM learns an adaptive weight for each client according to the data heterogeneity while FedAvg assigns a fixed weight for each client based on the number of local samples.

However, they inherently depend on label information during training, indicating that they adhere to supervised or semi-supervised learning paradigms. In practice, data labels are frequently inaccessible, and the lack of consistent semantics across clients further exacerbates the difficulties of federated learning (FL) [20, 21, 22, 23, 24, 25]. As a result, enabling effective FGL in an unsupervised setting poses a significant challenge. For unsupervised FGL, one of the most straightforward solutions is to leverage mature FL frameworks in conjunction with self-supervised learning (SSL) strategies [26, 27, 28, 29]. For example, clients perform local training using the SimCLR [30], while the server aggregates model parameters by FedAvg [31]. However, naively transplanting existing approaches fails to effectively cope with the common issue of heterogeneous data in FL. Moreover, some unsupervised FL algorithms originally designed for non-graph data are also worth mentioning. Studies [32, 33] used the clustering strategies to enhance the consensus of global space. Zhuang et al. [34, 35] proposed to dynamically update the local models based on the parameter differences between the global and local models. Liao et al. [36] addressed the problem of representation collapse in unsupervised FL.

These methods are insightful for unsupervised FL, but they exhibit two primary limitations: ❶ *Inability to align the representation spaces of various clients.* Due to the absence of labels, there are no consistent signals among clients, which can easily lead to shifts in the representation spaces and weaken the generalization for global model. Although studies [32, 36] adopt clustering or optimal transport (OT) to enhance the separability of representations, they merely alleviate representation collapse and do not achieve a unified representation space across clients. Especially, when faced with graph data, their learned representation spaces are prone to fall into suboptimality due to complex topological networks. ❷ *Unable to adaptively learn the optimal parameters of global model.* Most existing federated aggregation methods use weighted averaging based on number of local samples, which might degrade the performance of global model. Studies [34, 35, 36] introduce divergence-aware and multi-objective aggregation strategies, but their matrix-level designs fall short in effectively capturing the complementary information among local models. From Fig. 1(a), we can see that the scatter output by FedAvg is dispersed throughout the space, the nodes of the same categories from various clients cannot be clustered together, indicating that representation spaces of different clients are unaligned. The subgraph statistics for different clients are reported in Fig. 2(a). The total numbers of nodes in the different clients are almost the same, but the numbers of nodes in each category vary considerably, showing different heterogeneous situations. If the traditional FedAvg is used, each client is assigned to almost a same weight as shown in Fig. 2(b), which is clearly unreasonable.

In light of the above concerns, we propose a tailored framework FedPAM for unsupervised FGL, which mainly consists of two key modules: Representation Space Alignment (**RSA**) and Adaptive Global Parameter Learning (**AGPL**). Specifically, RSA aims to learn a set of cross-client anchors, which are used to define the global representation space. To align the representation spaces of various clients, the fused Gromov-Wasserstein optimal transport (FGW-OT) [37] is employed to establish mapping between local subgraphs and the anchor graph with minimal cost. Additionally, the anchors also serve as global space projector, projecting local embeddings into the global space to facilitate the contrastive learning, thereby mitigating the biased training caused by data heterogeneity. AGPL departs entirely from the conventional weighted aggregation paradigm. It stacks parameters of local models into third-order tensors, and leverages the low-rank tensor decomposition to capture high-order correlations among clients. Further, the optimal parameters of global models are adaptively learned in the low-rank tensor space, enabling effective integration of local knowledge. From Figs. 1(b) and 2(b), it can be observed that the proposed FedPAM can effectively align the representation spaces

across clients and adaptively learn the global model parameters. The framework of the proposed FedPAM is presented in Fig. 3. We summarize the contributions of this paper from four-fold:

- We are the first to identify the challenges inherent in unsupervised FGL and design a unified framework to address them. This framework proposes the representation space alignment and the adaptive global parameter learning for unsupervised FGL.

- We learn a set of anchors to span the global representation space, and adopt the FGW-OT to match local subgraphs with the anchor graph, thus achieving the alignment of representation spaces across multiple clients.

- We stack the local model parameters into third-order tensors and extract its low-rank components to capture the high-order correlations among clients. In the high-dimensional space, the optimal global model parameters are obtained by adaptive fusion.

- To verify the effectiveness of the proposed FedPAM, we conduct the comparative experiments on eight graph datasets. Compared to traditional and SOTA methods, FedPAM achieves more superior performance.

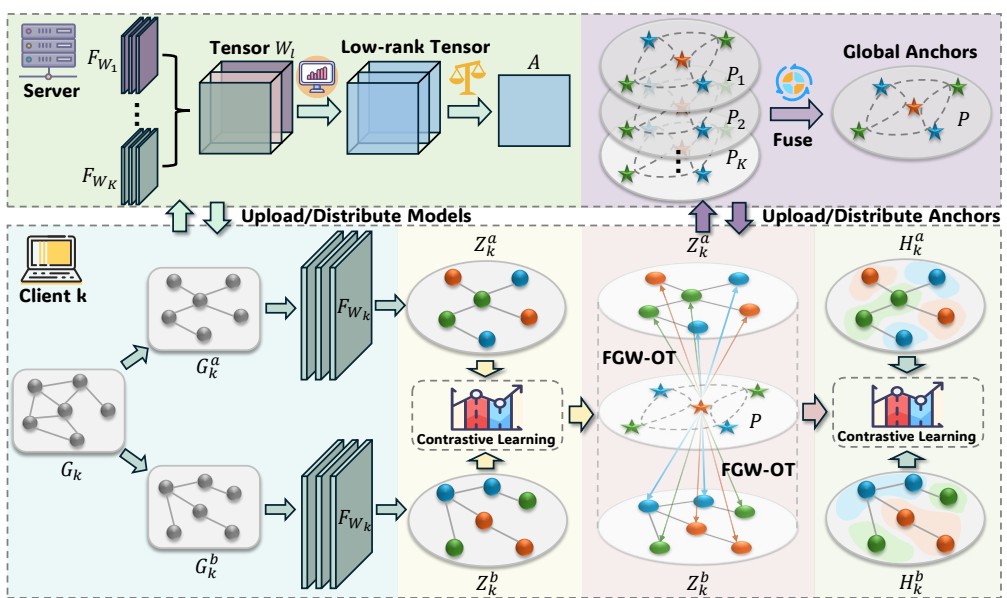

Figure 3: The framework of the proposed FedPAM. The client performs the RSA module and the server performs the AGPL module, models and anchors are transferred between the server and clients.

## 2 Preliminaries

**Federated Graph Learning**. Given a FGL system with a central server and $K$ clients, each client possesses a private subgraph $\mathcal{G}_k = \{\mathcal{V}_k, \mathcal{E}_k, \mathbf{X}_k\}$, where $\mathcal{V}_k$, $\mathcal{E}_k$, and $\mathbf{X}_k$ indicate the vector set, the edge set, and the feature matrix, respectively. Specifically, $\mathbf{X}_k \in \mathbb{R}^{N_k \times d}$ has $N_k$ nodes and $d$ dimensions. The adjacency matrix $\mathbf{A}_k \in \mathbb{R}^{N_k \times N_k}$ is induced by the edge set $\mathcal{E}_k$. If there is an edge between the $i$-th node and the $j$-th node, then $\mathbf{A}_k(i,j) = 1$. Otherwise, $\mathbf{A}_k(i,j) = 0$. In the unsupervised FGL, the node labels $\mathbf{Y}_k = [y_k^1, ..., y_k^{N_k}]$ are all unavailable during the GNN's training. Notably, we assume that all subgraphs originate from a complete graph, namely $\mathcal{G}_k \in \mathcal{G}$. At the beginning of a communication round in FL, the central server distributes the global GNN $F_{\mathbf{W}}$ parameterized by $\mathbf{W}$ to each client and starts the local training, the computation of a GNN can be summarized as

$$\mathbf{z}_i^{l+1} = \delta \left( F_{\mathbf{W}_l}(\mathbf{z}_i^l, \text{AGG}(\mathbf{z}_j^l, \mathbf{A}_{ij})) \right) |\forall v_j \in \mathcal{N}(v_i)), \tag{1}$$

where $\delta(\cdot)$ denotes the activation function, $F_{\mathbf{W}_l}$ denotes the $l$-th encoder layer, $\mathbf{z}_i^{l+1}$ is the $i$-th node's representation at the $(l+1)$-th layer, $\mathcal{N}(v_i)$ is the neighborhood node set for node $v_i$, $\text{AGG}(\cdot)$ denotes

the embedding fusion manner. When the local training finishes, local GNNs are uploaded to the server for aggregating the global GNN, which is formulated as

$$\mathbf{W} = \sum_{k=1}^{K} \frac{N_k}{N} \mathbf{W}_k, \tag{2}$$

where $N = \sum_{k=1}^{K} N_k$ denotes the sum of local nodes.

**Tensor Nuclear Norm**. For a third-order tensor $\boldsymbol{\mathcal{X}} \in \mathbb{R}^{n_1 \times n_2 \times n_3}$, its the $j$-th frontal slice is denoted as $\boldsymbol{\mathcal{X}}^{(j)} \in \mathbb{R}^{n_1 \times n_2}$. The fast Fourier transformation (FFT) is used to transform a tensor into the frequency domain, which is formulated as $\hat{\boldsymbol{\mathcal{X}}} = \mathrm{fft}(\boldsymbol{\mathcal{X}}, [], 3)$, its inverse operation is written as $\boldsymbol{\mathcal{X}} = \mathrm{ifft}(\hat{\boldsymbol{\mathcal{X}}}, [], 3)$. The product $\boldsymbol{\mathcal{Z}} \in \mathbb{R}^{n_1 \times n_4 \times n_3}$ of $\boldsymbol{\mathcal{X}} \in \mathbb{R}^{n_1 \times n_2 \times n_3}$ and $\boldsymbol{\mathcal{Y}} \in \mathbb{R}^{n_2 \times n_4 \times n_3}$ is defined as

$$\boldsymbol{\mathcal{Z}} = \boldsymbol{\mathcal{X}} * \boldsymbol{\mathcal{Y}} = \mathrm{fold}(\mathrm{bcirc}(\boldsymbol{\mathcal{X}}) \, \mathrm{bvec}(\boldsymbol{\mathcal{Y}})), \tag{3}$$

where $\mathrm{bvec}(\boldsymbol{\mathcal{Y}}) = [\boldsymbol{\mathcal{Y}}^{(1)}; ...; \boldsymbol{\mathcal{Y}}^{(n_3)}]$ is the block vectorizing operation, and $\mathrm{fold}(\mathrm{bvec}(\boldsymbol{\mathcal{Y}}))$ is the inverse operation. $\mathrm{bcirc}(\boldsymbol{\mathcal{X}})$ is defined as

$$bcirc(\boldsymbol{\mathcal{X}}) = \begin{bmatrix} \boldsymbol{\mathcal{X}}^{(1)} & \boldsymbol{\mathcal{X}}^{(n_3)} & \cdots & \boldsymbol{\mathcal{X}}^{(2)} \\ \boldsymbol{\mathcal{X}}^{(2)} & \boldsymbol{\mathcal{X}}^{(1)} & \cdots & \boldsymbol{\mathcal{X}}^{(3)} \\ \vdots & \vdots & \ddots & \vdots \\ \boldsymbol{\mathcal{X}}^{(n_3)} & \boldsymbol{\mathcal{X}}^{(n_3-1)} & \cdots & \boldsymbol{\mathcal{X}}^{(1)} \end{bmatrix}. \tag{4}$$

Further, if $\boldsymbol{\mathcal{X}}^T * \boldsymbol{\mathcal{X}} = \boldsymbol{\mathcal{X}} * \boldsymbol{\mathcal{X}}^T = \boldsymbol{\mathcal{I}}$, $\boldsymbol{\mathcal{X}}$ is called orthogonal tensor. $\boldsymbol{\mathcal{I}} \in \mathbb{R}^{n_1 \times n_1 \times n_3}$ is an identity tensor, where the first frontal slice is an identity matrix and the remaining slices are all zeros. The tensor singular value decomposition (T-SVD) is a key operation for calculating the tensor nuclear norm, their definitions are introduced as follows.

**Definition 1.** *For a tensor $\boldsymbol{\mathcal{X}} \in \mathbb{R}^{n_1 \times n_2 \times n_3}$, the T-SVD is formulated as*

$$\boldsymbol{\mathcal{X}} = \boldsymbol{\mathcal{U}} * \boldsymbol{\mathcal{D}} * \boldsymbol{\mathcal{V}}^T \tag{5}$$

*where $\boldsymbol{\mathcal{U}} \in \mathbb{R}^{n_1 \times n_1 \times n_3}$ and $\boldsymbol{\mathcal{V}} \in \mathbb{R}^{n_2 \times n_2 \times n_3}$ are orthogonal, $\boldsymbol{\mathcal{D}} \in \mathbb{R}^{n_1 \times n_2 \times n_3}$ is f-diagonal.*

**Definition 2.** *For a tensor $\boldsymbol{\mathcal{X}} \in \mathbb{R}^{n_1 \times n_2 \times n_3}$, its T-SVD based tensor nuclear norm [38] is defined as*

$$||\boldsymbol{\mathcal{X}}||_{\circledast} = \sum_{j=1}^{n_3} ||\hat{\boldsymbol{\mathcal{X}}}^{(j)}||_* = \sum_{i}^{\min(n_1,n_2)} \sum_{j=1}^{n_3} |\hat{\boldsymbol{\mathcal{D}}}^{(j)}(i,i)|, \tag{6}$$

*where $\hat{\boldsymbol{\mathcal{D}}}^{(j)}(i,i)$ is obtained by $\hat{\boldsymbol{\mathcal{X}}}^{(j)} = \hat{\boldsymbol{\mathcal{U}}}^{(j)} * \hat{\boldsymbol{\mathcal{D}}}^{(j)} * \hat{\boldsymbol{\mathcal{V}}}^{(j)^T}$.*

## 3 Methodology

In this section, we elaborate on the proposed FedPAM. Overall, FedPAM is comprised of two modules: **RSA** and **AGPL**. Specifically, **RSA** is used to align the representation spaces across clients through the FGW-OT between local subgraphs and anchor graph. **AGPL** is employed to adaptively learn the global model parameters in the low-rank tensor space.

### 3.1 Fused Gromov-Wasserstein Optimal Transport for Aligning Representation Spaces

**Local Self-Supervised Learning**. Self-supervised learning (SSL) is an effective technique for preventing representation collapse and enhancing the discriminative quality of learned representations. In the absence of node labels, local clients adopt a self-supervised strategy for training. For the $k$-th local subgraph $\mathcal{G}_k$, it is augmented to $\mathcal{G}_k^a$ and $\mathcal{G}_k^b$ via two different kinds of augment methods. Then, the local GNN $F_{\mathbf{W}_k}^k$ encodes them into the embedding space:

$$\mathbf{Z}_k^a = F_{\mathbf{W}_k}^k(\mathcal{G}_k^a), \mathbf{Z}_k^b = F_{\mathbf{W}_k}^k(\mathcal{G}_k^b). \tag{7}$$

A SSL loss between $\mathbf{Z}_k^a$ and $\mathbf{Z}_k^b$ is used for back propagation and generally written as

$$\mathcal{L}_{SSL} = \mathbb{E}_{\forall i,j \in [N_k]} \ell_{pos}(\mathbf{z}_i^a, \mathbf{z}_i^b) - \lambda \ell_{neg}(\mathbf{z}_i^a, \{\mathbf{z}_j^b\}_{i \neq j}), \tag{8}$$

where $\ell_{pos}$ and $\ell_{neg}$ denote the losses for positive and negative pairs, respectively. Overall, the SSL loss $\mathcal{L}_{SSL}$ can be categorized into two classes: contrastive and non-contrastive. Contrastive methods require the involvement of negative samples during the computation of SSL, while non-contrastive methods do not rely on negative samples. We investigate the impacts of various SSL on the proposed framework in the experimental section.

**FGW-OT between Local Graphs and Anchor Graph**. Under our assumption, local subgraphs originate from a underlying global graph, implying that node representations across clients following a unified distribution. However, due to the heterogeneous nature of local data, the embeddings learned by local clients tend to exhibit distributional discrepancies. Moreover, since data from other clients is unaccessible, local models lack awareness of the global distribution. To address this issue, we introduce $N_P$ learnable anchors $\mathbf{P} = \{\mathbf{p}_1, ..., \mathbf{p}_{N_P}\}$ in the server to span the global space, the relationships between anchors are recorded by the anchor graph $\mathbf{A}_P$. These anchors also serve as the global projector, enabling the projection of local embedding into the global space. At the beginning of each communication round, the global model and the anchors are distributed to each client for local training. To align local subgraphs with anchor graph, the FGW-OT is introduced, whose definition is written as

**Definition 3.** *Given two metric measure space $\mathcal{G}_1 = (\mathcal{V}_1, \mathbf{A}_1, \mu_1)$ and $\mathcal{G}_1 = (\mathcal{V}_2, \mathbf{A}_2, \mu_2)$, $\mathcal{V}_1$ and $\mathcal{V}_2$ denote the node sets, where the data sizes are respectively $N_1$ and $N_2$, $\mathbf{A}_1$ and $\mathbf{A}_2$ denote the relationship matrices, $\mu_1$ and $\mu_2$ denote the data marginal distributions. Then, the FGW-OT distance between $\mathcal{G}_1$ and $\mathcal{G}_2$ is defined as*

$$\inf_{\pi \in \Pi} \sum_{i,j}^{N_1} \sum_{m,n}^{N_2} ((1-\alpha) D(\mathbf{x}_{1,i}, \mathbf{x}_{2,j})^p + \alpha |[\mathbf{A}_1]_{i,j} - [\mathbf{A}_2]_{m,n}|^p) \pi_{i,m} \pi_{j,n} \tag{9}$$

$$s.t. \ \Pi = \{\pi \in \mathbb{R}_+^{N_1 \times N_2} | \pi \mathbf{1}_{N_2} = \mu_1, \pi^T \mathbf{1}_{N_1} = \mu_2\}$$

*where $\mathbf{x}_{1,i} \in \mathcal{V}_1$, $\mathbf{x}_{2,j} \in \mathcal{V}_2$, $D(\cdot)$ measures the distance between two nodes, $\alpha$ is a trade-off parameter between the feature-level distance and the relationship-level distance, $p$ is a constant used to adjust the strength of the distance, $\mathbf{1}_{N_1}$ and $\mathbf{1}_{N_2}$ denote the two vectors of all-one elements with $N_1$ and $N_2$ dimensions. $\pi \in \mathbb{R}^{N_1 \times N_2}$ is the expected optimal transport plan.*

In context of graph learning, FGW-OT can measure the sum of feature and topology distances between two graphs. Since anchors are derived from the global representation space, thus the anchor graph can be optimally transported to each local subgraph $\mathcal{G}_k$ with minimal cost. The optimization objective is formulated as

$$\min_{\pi_k \in \Pi} \sum_{i,j}^{N_k} \sum_{m,n}^{N_P} \left( (1-\alpha)(-[\mathbf{P}_k]_m [\mathbf{Z}_k^T]_i) + \alpha |[\mathbf{A}_k]_{i,j} - [\mathbf{A}_P]_{m,n}| \right) [\pi_k]_{i,m} [\pi_k]_{j,n} - \epsilon \, \mathrm{H}(\pi_k), \tag{10}$$

$$\text{s.t. } \Pi = \{\pi_k \in \mathbb{R}_+^{N_k \times N_P} | \pi_k \mathbf{1}_{N_P} = \mu_1, \pi_k^T \mathbf{1}_{N_k} = \mu_2\},$$

where $\mathrm{H}(\cdot)$ denotes the entropy of certain variable, $\epsilon$ denotes a trade-off constant. The entropy regularization is introduced to ensure that the optimization problem has a unique solution. The product $\mathbf{P}_k \mathbf{Z}_k^T$ measures the similarity between the anchors and nodes, then $-\mathbf{P}_k \mathbf{Z}_k^T$ is the distance between the two terms. Problem (10) can be optimized by the proposed method in [37], then an optimal transport matrix $\pi_k \in \mathbb{R}^{N_k \times N_P}$ is obtained. $\pi_k$ reflects the unnormalized transfer probability from nodes to anchors, and its normalized form is formulated as

$$[\mathbf{T}_k]_{i,j} = \frac{[\pi_k]_{i,j}}{\sum_{j'} [\pi_k]_{i,j'}}. \tag{11}$$

For the anchor-node similarity matrix $\mathbf{P}_k \mathbf{Z}_k^T$, its has two-fold meanings. **First**, it depicts the relationships between anchors and nodes, and its normalized form can also be viewed as their transfer probability. Thus, we have

$$[\mathbf{C}_k]_{i,j} = \frac{\exp([\mathbf{Z}_k]_i [\mathbf{P}_k^T]_j / \tau)}{\sum_{j'} \exp([\mathbf{Z}_k]_i [\mathbf{P}_k^T]_{j'} / \tau)}. \tag{12}$$

$\mathbf{T}_k$ is closed-form solution obtained through algebraic computation and does not have gradients, whereas $\mathbf{C}_k$ is an inexact solution with gradients. In addition, $\mathbf{C}_k$ is also used to conduct the anchor

graph $\mathbf{A}_P = \mathbf{C}_k^T \mathbf{C}_k$. $\mathbf{C}_k$ is expected to drive close to $\mathbf{T}_k$, then the loss is written as:

$$\mathcal{L}_{CE} = -\sum_{i=1}^{N_k} \sum_{j=1}^{N_P} \left([\mathbf{T}_k]_{i,j} \log[\mathbf{C}_k]_{i,j}\right). \tag{13}$$

Notably, as mentioned above, there are two augmented latent embeddings: $\mathbf{Z}_k^a$ and $\mathbf{Z}_k^b$. We perform the OT mapping on the two views, then $\mathcal{L}_{CE}$ is reformulated as

$$\mathcal{L}_{CE} = \mathcal{L}_{CE}^a + \mathcal{L}_{CE}^b. \tag{14}$$

**Second**, the anchor matrix $\mathbf{P}$ represents the entire representation space and is regarded as a global space projector. From another perspective, $\mathbf{P}\mathbf{Z}^T$ is understood as a global-aware representation. In Eq. (8), it performs the contrastive learning with local representations and is denoted as $\mathcal{L}_{SSL}^l$. In addition, we expect to the global-aware representations to be more discriminative as well. By projecting $\mathbf{Z}_k^a$ and $\mathbf{Z}_k^b$ into global space via anchors $\mathbf{P}_k$, the global-aware representations $\mathbf{H}_k^a = \mathbf{Z}_k^a \mathbf{P}_k^T$ and $\mathbf{H}_k^b = \mathbf{Z}_k^b \mathbf{P}_k^T$ are obtained, then the contrastive learning between them is executed through Eq. (8), the loss is denoted as $\mathcal{L}_{SSL}^g$. Thus, the local SSL loss is reformulated as

$$\mathcal{L}_{SSL} = \mathcal{L}_{SSL}^l + \mathcal{L}_{SSL}^g. \tag{15}$$

In summary, the training loss in local clients is concluded as

$$\mathcal{L} = \mathcal{L}_{SSL} + \lambda \mathcal{L}_{CE}, \tag{16}$$

where $\lambda$ denotes the trade-off parameter. Through the back propagation from loss $\mathcal{L}$, the local GNNs and anchors are optimized. When the local training completes, the local GNNs and anchors are uploaded to sever for aggregation.

## 3.2 Low-Rank Tensor Optimization for Adaptively Learning Global GNN Parameters

Traditional federated aggregation performs weighted parameter fusion based on number of local samples. However, this approach has two drawbacks. First, the matrix-level fusion cannot capture the high-order correlations between clients. Second, the fixed weights cannot adaptively determine the importance of each client. In light of the shortcomings, we propose to find the optimal global GNN's parameters in a low-rank tensor space. Concretely, when the server receives local GNNs' parameters, the $l$-th layer's parameters $\{\mathbf{W}_{k,l}\}_{k=1}^K$ are stacked as a third-order tensor $\mathcal{W}_l$. We argue that the low-rank component $\mathcal{L}_l$ in $\mathcal{W}_l$ contains the globally shared information, and the noise component $\mathcal{E}_l$ contains the client-specific information. Then, the former needs to be retained, while the latter should be discarded. Further, we aim to adaptively fuse local GNNs' parameters in the low-rank tensor space for obtaining the optimal global GNN parameters $\mathbf{A}_l$. Considering the above concerns, the following optimization objective is proposed (the subscript $l$ is omitted for a concise expression):

$$\min_{\mathcal{L}, \mathcal{E}, \mathbf{A}, \alpha_k} \|\mathcal{L}\|_{\circledast} + \beta\|\mathcal{E}\|_1 + \sum_{k=1}^K (\alpha_k)^r \left(\|\mathbf{L}_k - \mathbf{A}\|_F^2\right)$$

$$\text{s.t. } \mathcal{W} = \mathcal{L} + \mathcal{E}, \sum_{k=1}^K \alpha_k = 1, \alpha_k \geq 0, \forall k \in [K], \tag{17}$$

where $\beta$ is a trade-off parameter, $\alpha_k$ is the weight for the $k$-th client, $r$ is a constant to smooth the distribution of weights. The optimization problem (17) can be solved by the alternating direction method of multipliers (ADMM). The augmented Lagrangian function of (17) is conducted as follows:

$$\mathcal{F}\left(\mathcal{L}; \mathcal{E}; \mathcal{G}; \mathbf{A}; \{\alpha_k\}_{k=1}^K\right)$$

$$= \|\mathcal{G}\|_{\circledast} + \beta\|\mathcal{E}\|_1 + \sum_{k=1}^K (\alpha_k)^r \|\mathbf{L}_k - \mathbf{A}\|_F^2 + \langle \mathcal{Y}, \mathcal{W} - \mathcal{L} - \mathcal{E} \rangle + \frac{\mu}{2}\|\mathcal{W} - \mathcal{L} - \mathcal{E}\|_F^2 \tag{18}$$

$$+ \langle \mathcal{H}, \mathcal{G} - \mathcal{L} \rangle + \frac{\mu}{2}\|\mathcal{G} - \mathcal{L}\|_F^2,$$

where $\mathcal{G}$ is an auxiliary variable, $\mathcal{Y}$ and $\mathcal{H}$ denote the two Lagrange multipliers, $\mu$ is a trade-off parameter. The update rule for each variable is presented as follows.

**Update $\mathcal{G}$**: Fixing $\mathcal{L}$, $\mathcal{E}$, $\mathbf{A}$, $\{\alpha_k\}_{k=1}^{K}$, the subproblem with respect to $\mathcal{G}$ is written as

$$\min_{\mathcal{G}} \|\mathcal{G}\|_{\circledast} + \langle \mathcal{H}, \mathcal{G} - \mathcal{L} \rangle + \frac{\mu}{2}\|\mathcal{G} - \mathcal{L}\|_F^2$$
$$= \min_{\mathcal{G}} \|\mathcal{G}\|_{\circledast} + \frac{\mu}{2}\left\|\mathcal{G} - \left(\mathcal{L} - \frac{1}{\mu}\mathcal{H}\right)\right\|_F^2. \tag{19}$$

For the solution of low-rank tensor $\mathcal{G}$, study [39] provided a tensor tubal-shrinkage method $\mathcal{R}$, the updated $\mathcal{G}^*$ is obtained by

$$\mathcal{G}^* = \mathcal{R}_{1/\mu}(\mathcal{L} - \frac{1}{\mu}\mathcal{H}). \tag{20}$$

**Update $\mathcal{L}$**: When optimizing $\mathcal{L}$, the update rule for its each slice $\mathbf{L}_k$ is same. Fixing $\mathcal{G}$, $\mathcal{E}$, $\mathbf{A}$, $\{\alpha_k\}_{k=1}^{K}$, the subproblem for $\mathbf{L}_k$ is formulated as

$$\min_{\mathbf{L}_k} (\alpha_k)^r \|\mathbf{L}_k - \mathbf{A}\|_F^2 + \langle \mathbf{Y}_k, \mathbf{W}_k - \mathbf{L}_k - \mathbf{E}_k \rangle + \frac{\mu}{2}\|\mathbf{W}_k - \mathbf{L}_k - \mathbf{E}_k\|_F^2$$
$$+ \langle \mathbf{H}_k, \mathbf{G}_k - \mathbf{L}_k \rangle + \frac{\mu}{2}\|\mathbf{G}_k - \mathbf{L}_k\|_F^2. \tag{21}$$

Taking the partial derivative with respect to $\mathbf{L}_k$ and setting it to zero, the update rule for $\mathbf{L}_k^*$ is derived:

$$\mathbf{L}_k^* = \frac{2(\alpha_k)^r\mathbf{A} + \mu\mathbf{Z}_k - \mu\mathbf{E}_k + \mathbf{Y}_k + \mu\mathbf{G}_k + \mathbf{H}_k}{2(\alpha_k)^r + 2\mu} \tag{22}$$

**Update $\mathbf{A}$**: Masking variables unrelated to $\mathbf{A}$, we obtain the following subproblem:

$$\min_{\mathbf{A}} \sum_{k=1}^{K}(\alpha_k)^r(\|\mathbf{L}_k - \mathbf{A}\|_F^2). \tag{23}$$

The update rule for $\mathbf{A}^*$ can be obtained by computing the partial derivative with respect to $\mathbf{A}^*$ and equating it to zero, then we have

$$\mathbf{A}^* = \frac{\sum_{k=1}^{K}(\alpha_k)^r\mathbf{L}_k}{\sum_{k=1}^{K}(\alpha_k)^r}. \tag{24}$$

**Update $\mathcal{E}$**: When $\mathcal{E}$ is updated, the other variables are viewed as constants, the subproblem is formulated as

$$\min_{\mathcal{E}} \beta\|\mathcal{E}\|_1 + \langle \mathcal{Y}, \mathcal{W} - \mathcal{L} - \mathcal{E} \rangle + \frac{\mu}{2}\|\mathcal{W} - \mathcal{L} - \mathcal{E}\|_F^2$$
$$= \min_{\mathcal{E}} \beta\|\mathcal{E}\|_1 + \frac{\mu}{2}\|\mathcal{E} - (\mathcal{W} - \mathcal{L} + \mathcal{Y}/\mu)\|_F^2. \tag{25}$$

Let $\mathcal{B} = \mathcal{W} - \mathcal{L} + \mathcal{Y}/\mu$, the update method with respect to $\mathcal{E}$ is

$$\mathcal{E}^* = \max(\mathcal{B} - \lambda/\mu, 0) + \min(\mathcal{B} + \lambda/\mu, 0). \tag{26}$$

**Update $\alpha_k$**: Neglecting variables unrelated to $\alpha_k$, the subproblem about $\alpha_k$ is written as

$$\min_{\alpha_k} \sum_{k=1}^{K}(\alpha_k)^r(\|\mathbf{L}_k - \mathbf{A}\|_F^2), \text{s.t. } \sum_{k=1}^{K}\alpha_k = 1, \alpha_k \geq 0, \forall k \in [K]. \tag{27}$$

Taking the partial derivative with respect to $\alpha_k$ and setting it to zero, we have

$$\alpha_k^* = \frac{(\|\mathbf{L}_k - \mathbf{A}\|_F^2)^{1/(1-r)}}{\sum_{k=1}^{K}(\|\mathbf{L}_k - \mathbf{A}\|_F^2)^{1/(1-r)}}. \tag{28}$$

**Update $\mathcal{Y}$, $\mathcal{H}$, $\mu$**:

$$\mathcal{H}^* = \mathcal{H} + \mu(\mathcal{G} - \mathcal{L}); \mathcal{Y}^* = \mathcal{Y} + \mu(\mathcal{W} - \mathcal{L} - \mathcal{E}); \mu^* = \min(\omega * \mu, \mu_{max}), \tag{29}$$

where $\omega$ and $\mu_{max}$ are two predefined parameters. The optimal global GNN parameters are obtained through the iterative optimization based on ADMM described above. Thus, the global GNN parameters and anchors are distributed to clients in the next communication.

Table 1: Performance comparison on Cora, CiteSeer, PubMed, and Ogbn-Arxiv datasets with three different SSL losses, where the optimal results are **bolded** and the suboptimal results are underlined.

| SSL | Method | Cora | | CiteSeer | | PubMed | | Ogbn-Arxiv | |
|---|---|---|---|---|---|---|---|---|---|
| | | ACC | Fscore | ACC | Fscore | ACC | Fscore | ACC | Fscore |
| Simsiam | FedAvg | 54.38 | 39.27 | 36.30 | 20.34 | 63.81 | 50.88 | 35.26 | 20.49 |
| | FedProx | 55.35 | 40.83 | 36.80 | 21.34 | 63.81 | 50.88 | 34.61 | 20.03 |
| | MOON | 54.03 | 39.81 | 35.94 | 20.41 | 63.80 | 50.87 | 35.46 | 21.20 |
| | FedU$^2$ | 56.04 | 42.65 | 39.00 | 25.77 | 61.27 | 48.09 | 35.39 | 22.00 |
| | FedPAM | **61.40** | **52.85** | **49.53** | **38.73** | **64.23** | **51.92** | **35.97** | **23.07** |
| SimCLR | FedAvg | 54.65 | 39.95 | 35.48 | 19.62 | 64.08 | 51.46 | 45.58 | 33.08 |
| | FedProx | 55.26 | 40.68 | 35.56 | 19.78 | 63.95 | 51.17 | 44.40 | 31.86 |
| | MOON | 54.56 | 39.81 | 36.30 | 20.34 | 64.42 | 52.17 | 46.23 | 33.96 |
| | FedX | 56.73 | 43.11 | 37.10 | 22.37 | 61.77 | 49.09 | 46.69 | 34.77 |
| | FedU$^2$ | 57.09 | 43.45 | 40.89 | 27.54 | 66.79 | 56.25 | **48.41** | 36.46 |
| | FedPAM | **58.62** | **45.21** | **43.90** | **31.61** | **69.22** | **59.42** | 48.19 | **37.25** |
| BYOL | FedAvg | 63.01 | 50.51 | 42.65 | 29.72 | 61.85 | 49.62 | 35.29 | 20.73 |
| | FedU | 64.06 | 51.92 | 46.81 | 34.98 | 62.08 | 49.82 | 38.07 | 24.15 |
| | FedEMA | 63.53 | 51.23 | 43.59 | 30.97 | 64.47 | 53.39 | 38.12 | 24.07 |
| | Orchestra | 54.47 | 39.63 | 37.54 | 24.39 | 61.27 | 48.09 | 35.26 | 20.49 |
| | FedU$^2$ | 65.11 | 54.16 | 47.32 | 35.79 | 66.51 | 55.86 | 39.95 | 26.03 |
| | FedPAM | **66.01** | **55.23** | **48.81** | **38.69** | **66.67** | **56.25** | **40.55** | **26.79** |

## 4 Experiments

### 4.1 Experimental Setups

**Graph Datasets**. Eight graph datasets are selected as benchmark datasets, including **Cora**, **CiteSeer**, **PubMed**, **Ogbn-Arxiv**, **Computers**, **Photo**, **Physics**, **Amazon-ratings**. The eight datasets vary in terms of scales, and cover different types, such as citation network, co-purchase network, co-author network, rating network. Each graph is divided into ten subgraphs via the Louvain method [40].

**Compared Methods**. We compare the proposed FedPAM with typical and SOTA FL algorithms, including **FedAvg** [31], **FedProx** [41], **MOON** [42], **FedX** [43], **FedU** [34], **FedEMA** [35], **Orchestra** [32], **FedU$^2$** [36]. In local training, three SSL strategies are adopted, including **Simsiam** [44], **SimCLR** [30], **BYOL** [45]. Specifically, **Simsiam** and **BYOL** are non-contrastive methods while **SimCLR** is contrastive method. We adopt the standard linear probing to evaluate the performance of algorithms, ACC and Fscore are used as the evaluation metrics.

### 4.2 Performance Comparison

We evaluate the performance of FL algorithms using three SSL strategies on eight graph datasets, the results are reported in Tables 1 and 2. Some interesting phenomenon can be observed. First, FL algorithms exhibit varying performance when equipped with different SSL losses. For instance, on Cora dataset, employing BYOL leads to improvements by 8.63% and 8.36% in ACC compared to the other two SSL losses for FedAvg, respectively. Encouragingly, the proposed FedPAM consistently achieves superior performance with different SSL methods, validating the contributions of RSA and AGPL. Second, compared to baseline methods, the proposed FedPAM demonstrates more significant improvements on homogeneous graphs than on heterogeneous graphs. Node attributes and edge types in homogeneous graphs are less diverse, then the learnable anchors are more likely to span the global representation space. Furthermore, homogeneous graphs induce local models to be more similar, facilitating low-rank tensor optimization to find better global model parameters.

### 4.3 Ablation Study

RSA and AGPL paly critical roles for the proposed FedPAM. The results of their ablation studies are reported in Table 3. When both modules are removed, the model degenerates into FedAvg and

Table 2: Performance comparison on Computers, Photo, Physics, and Amazon-ratings datasets with three different SSL losses, where the optimal results are **bolded** and the suboptimal results are underlined.

| SSL | Method | Computers | | Photo | | Physics | | Amazon-ratings | |
|---|---|---|---|---|---|---|---|---|---|
| | | ACC | Fscore | ACC | Fscore | ACC | Fscore | ACC | Fscore |
| Simsiam | FedAvg | 63.09 | 50.92 | 72.01 | 63.56 | 75.55 | 67.73 | 36.72 | 19.79 |
| | FedProx | 64.05 | 51.70 | 73.26 | 65.08 | 75.33 | 66.22 | 36.64 | 19.83 |
| | MOON | 63.92 | 52.16 | 72.46 | 64.20 | 77.82 | 70.79 | 36.69 | 19.86 |
| | FedU$^2$ | 63.20 | 51.03 | 72.65 | 64.57 | 86.26 | 81.02 | 36.72 | 19.79 |
| | FedPAM | **64.19** | **53.93** | **74.06** | **66.96** | **86.51** | **81.86** | **37.45** | **24.97** |
| SimCLR | FedAvg | 78.21 | 72.27 | 84.59 | 79.82 | 79.07 | 71.30 | 36.56 | 20.06 |
| | FedProx | 78.41 | 72.56 | 84.66 | 79.82 | 77.05 | 67.74 | 36.46 | 19.92 |
| | MOON | 78.90 | 72.53 | 84.72 | 79.96 | 80.23 | 73.10 | 36.72 | 20.05 |
| | FedX | 80.60 | 75.09 | 85.49 | 80.72 | 80.31 | 73.60 | 36.70 | 19.79 |
| | FedU$^2$ | 79.24 | 73.34 | 84.94 | 80.15 | 81.03 | 74.27 | 39.58 | 27.92 |
| | FedPAM | **81.63** | **77.42** | **85.78** | **81.25** | **83.76** | **78.16** | **39.60** | **29.69** |
| BYOL | FedAvg | 63.38 | 52.20 | 71.12 | 61.93 | 84.99 | 79.13 | 36.77 | 21.05 |
| | FedU | 65.00 | 54.92 | 74.19 | 67.42 | 85.36 | 79.54 | 37.11 | 24.07 |
| | FedEMA | 65.44 | 56.38 | 76.81 | 70.31 | 83.39 | 77.53 | 38.32 | **27.22** |
| | Orchestra | 62.84 | 52.67 | 73.74 | 67.22 | 72.16 | 62.10 | 36.81 | 20.12 |
| | FedU$^2$ | 65.55 | 54.73 | 74.86 | 68.11 | 85.41 | 79.74 | 36.82 | 20.50 |
| | FedPAM | **66.66** | **57.57** | **80.50** | **75.34** | **86.43** | **80.98** | **38.43** | 26.52 |

Table 3: Ablation study with respect to two key modules **RSA** and **AGPL** on Cora and CiteSeer datasets, where SimCLR loss is used in the local training.

| RSA | AGPL | Cora | | CiteSeer | |
|---|---|---|---|---|---|
| | | ACC | Fscore | ACC | Fscore |
| ✗ | ✗ | 54.65 | 39.95 | 35.48 | 19.62 |
| ✗ | ✓ | 56.65 | 44.18 | 41.35 | 27.80 |
| ✓ | ✗ | 58.03 | **45.27** | 37.89 | 22.65 |
| ✓ | ✓ | **58.62** | 45.21 | **43.90** | **31.61** |

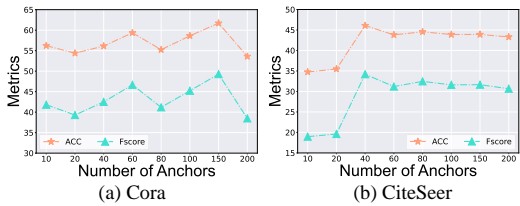

Figure 4: Performance comparison with different numbers of anchors on Cora and CiteSeer datasets, where SimCLR loss is used in the local training.

naturally achieves the worst performance. When either RSA and AGPL module is present, the performance of the model is improved, indicating that both representation space alignment and adaptive global parameter learning are effective. Certainly, the performance is optimal with both modules available.

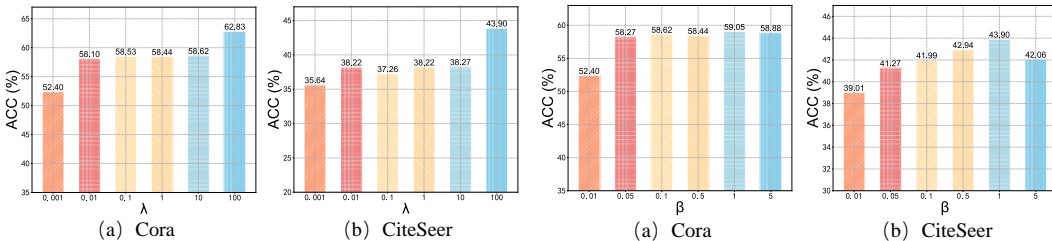

Figure 5: Sensitivity study with respect to $\lambda$ on Cora and CiteSeer datasets, where SimCLR loss is used in the local training.

Figure 6: Sensitivity study with respect to $\beta$ on Cora and CiteSeer datasets, where SimCLR loss is used in the local training.

## 4.4 Parameter Sensitivity Investigation

We investigate the impacts of three key hyperparameters in the proposed FedPAM, including the number of anchors $M$, $\lambda$ in the local training, and $\beta$ in the low-rank tensor optimization. From Fig. 4, it can be seen that either too few or too many anchors is detrimental to achieving optimal

performance. Excessive anchors tend to reduce discriminability, while too few anchors may lead to representation collapse. $\lambda$ controls the strength of the optimal transport, a large value of $\lambda$ facilitates better alignment of representation spaces as shown in Fig. 5. However, setting $\beta$ too high is not advisable as shown in Fig. 6, because it may hinder the capture of client-specific information.

## 5 Conclusion

In this paper, we address two key challenges in unsupervised FGL: the misalignment of representation spaces and non-adaptive federated aggregation schemas. For the first challenge, we propose the RSA module, which aims to learn a set of anchors across clients and use them to align representation spaces of different clients. For the second challenge, the AGPL module is introduced, employing the low-rank tensor optimization to adaptively learn a global model. Finally, the experimental results on eight graph datasets verify the effectiveness of the proposed FedPAM. More details about related work, algorithm, dadasets, and experimental results can be referred in the Appendix.

## Acknowledgments

The research is supported by the National Key R&D Program of China (2023YFB2703700), the National Natural Science Foundation of China (62176269).

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

# A   Summary of Appendix

We present the following sections in the appendix as supplement to this manuscript:

- Related work about FGL, OT, and low-rank tensor optimization.
- The algorithm flow of the proposed FedPAM.
- Implementation details for the proposed FedPAM.
- The analysis for the computational complexity.
- The details of the eight graph datasets used.
- The performance comparison using KNN as the evaluation metric.
- The performance comparison with large-scale clients.
- The performance comparison of plugging the AGPL module into existing works.

# B   Related Work

## B.1   Federated Graph Learning

The objective of federated graph learning is to enable the collaborative training of a GNN on multiple decentralized graphs, thereby improving its generalization. In terms of training data, FGL is generally divided into two categories: graph-level and node-level. Graph-level FGL focuses on addressing challenges related to label or feature heterogeneity. For example, [15, 16] proposed to cluster clients or share topology patterns for Non-IID graphs. To mitigate the domain gaps between private graphs, [46, 17] captured the cross-domain knowledge to enhance the local personalized inference. In contrast, node-level FGL primarily targets challenges arising from topological heterogeneity. [18, 9, 19] leveraged the global information to assist the local training. [47, 48] emphasized the impact of topological structures on federated learning and proposed the topology-aware aggregation method. While the aforementioned approaches provide valuable insights, they overlook the problem of FGL in unsupervised scenarios. We recognize the challenges of unsupervised FGL and propose a novel approach to address them.

## B.2   Optimal Transport

Optimal transport [49] is used to find the shortest distance for transforming one distribution into another, and has been widely applied in the fields of domain adaption and domain generalization. [50, 51] aligned the distributions between the source domain and the target domain via OT. As an extension of classical optimal transport, GW-OT is designed to align structural relationships between data distributions, making it particularly suitable for applications in graph match tasks. For instance, [52, 53] aimed to match the topological structures between various graphs via GW-OT. Furthermore, to simultaneously align the distributions in terms of features and structures, the FGW-OT is proposed. [37, 5] adopted the FGW-OT to achieve optimal mapping from source graphs to target graphs through features and topologies. For unsupervised FGL, how to explore the global representation space is critical challenge, we leverage the FGW-OT to learn a group of anchors across multiple private subgraphs for positioning the global coordinate.

## B.3   Low-rank Tensor Decomposition

Low-rank tensor decomposition seeks to recover the underlying low-rank components from a given tensor, and it has found extensive applications in image restoration, data mining, and data compression tasks. [54, 55, 56] employed the tensor singular value decomposition to explore the principal information of multi-source data. Deep learning provides a new solution for exploring the low-rankness, [57, 58] used neural networks to nonlinearly capture the low-rank structures. Although there have been some existing studies working on federated tensor computation [59, 60, 61], they mainly focused on distributed tensor decomposition with respect to data features. In contrast, we leverage low-rank tensor decomposition to exploit the complementarity of local models and solve for the global model parameters in the low-rank tensor space.

## C   The Algorithm Flow of the Proposed FedPAM

---

**Algorithm 1** The main steps of FedPAM

---

**Input:** Number of clients $K$, communication rounds $T$, local training epochs $E$, learning rate $\eta$, trade-off parameters $\lambda$ and $\beta$, local subgraph $\mathcal{G}_k = (\mathcal{V}_k, \mathcal{E}_k, \mathbf{X}_k)$, local GNN $F_{\mathbf{W}_k}$.
**Output:** Global model $F_{\mathbf{W}}$ and Global anchors $\mathbf{P}$.
1:   **Client Side:**
2:   **for** $k = 1 : K$ **in parallel do**
3:     **for** epoch $e = 1 : E$ **do**
4:       Augment the original local subgraph $\mathcal{G}_k$ into $\mathcal{G}_k^a$ and $\mathcal{G}_k^b$ via two different methods;
5:       Encode the two views of subgraphs: $\mathbf{Z}_k^a \leftarrow F_{\mathbf{W}_k}^k(\mathcal{G}_k^a)$, $\mathbf{Z}_k^b \leftarrow F_{\mathbf{W}_k}^k(\mathcal{G}_k^b)$;
6:       Calculate the SSL loss for local representations: $\mathcal{L}_{SSL}^l \leftarrow (\mathbf{Z}_k^a, \mathbf{Z}_k^b)$ by Eq. (8);
7:       **for** view $i$ in $(a, b)$ **do**
8:         Calculate the optimal transport matrix $\mathbf{T}_k^i$ between $\mathbf{Z}_k^i$ and $\mathbf{P}_k$ by Algorithm 2;
9:         Calculate the normalized transport matrix: $\mathbf{T}_k^i \leftarrow (\pi_k^i)$ by Eq. (11);
10:        Calculate the transfer probability matrix: $\mathbf{C}_k^i \leftarrow (\mathbf{Z}_k^i, \mathbf{P}_k)$ by Eq. (12);
11:        Calculate the CE loss $\mathcal{L}_{CE}^i \leftarrow (\mathbf{T}_k^i, \mathbf{C}_k^i)$ by (13);
12:       **end for**
13:       Calculate the CE loss for two views: $\mathcal{L}_{CE} \leftarrow (\mathcal{L}_{CE}^a, \mathcal{L}_{CE}^b)$;
14:       Obtain the global-aware representations: $\mathbf{H}_k^a \leftarrow \mathbf{Z}_k^a \mathbf{P}_k^T$, $\mathbf{H}_k^b \leftarrow \mathbf{Z}_k^b \mathbf{P}_k^T$;
15:       Calculate the SSL loss for global-aware representations: $\mathcal{L}_{SSL}^g \leftarrow (\mathbf{H}_k^a, \mathbf{H}_k^b)$ by Eq. (8);
16:       Calculate the total SSL loss $\mathcal{L}_{SSL} \leftarrow (\mathcal{L}_{SSL}^l, \mathcal{L}_{SSL}^g)$ by Eq. (15);
17:       Update local GNN $\mathbf{W}_k^e \leftarrow \mathbf{W}_k^{e-1} - \eta\nabla(\mathcal{L}_{SSL} + \mathcal{L}_{CE})$;
18:       Update local anchors $\mathbf{P}_k^e \leftarrow \mathbf{P}_k^{e-1} - \eta\nabla(\mathcal{L}_{SSL} + \mathcal{L}_{CE})$;
19:     **end for**
20:     Upload the local GNN $F_{\mathbf{W}_k}^k$ and local anchors $\mathbf{P}_k$ to the server;
21:   **end for**
22:   **Server Side:**
23:   **for** $t = 1 : T$ **do**
24:     **for** $i = 1 : L$ **do**
25:       Stack the $l$-th layer's parameters $\{\mathbf{W}_{k,l}\}$ of local model into a third-order tensor $\mathcal{W}_l$;
26:       Obtain the optimal parameters $\mathbf{A}_l$ from $\mathcal{W}_l$ by Algorithm 4;
27:     **end for**
28:     Obtain the global anchors $\mathbf{P} = \sum_{k=1}^K \frac{N_k}{N}\mathbf{P}_k$;
29:     Distribute the global GNN $F_{\mathbf{W}}$ and the global anchors $\mathbf{P}$ to clients;
30:   **end for**

---

**Algorithm 2** The main steps of fused GW-OT

---

**Input:** Node marginal distribution $\nu \in \mathbb{R}^N$, anchor marginal distribution $\mu \in \mathbb{R}^M$, adjacency matrix $\mathbf{A} \in \mathbb{R}^{N \times N}$, anchor adjacency matrix $\mathbf{S} \in \mathbb{R}^{M \times M}$, the transport matrix $\varphi \in \mathbb{R}^{N \times M}$ in terms of feature level, trade-off parameters $\alpha$ and $\epsilon$, maximum iteration round $T$.
**Output:** The transport matrix $\pi \in \mathbb{R}^{N \times M}$.
1:   $\pi \leftarrow \nu\mu^T$;
2:   $G_1 \leftarrow \mathbf{S} \odot \mathbf{S} \cdot \nu\mathbb{1}^T$;
3:   $G_2 \leftarrow \mathbb{1}(\mathbf{A} \odot \mathbf{A} \cdot \mu)^T$;
4:   **for** $t = 1 : T$ **do**
5:     $G \leftarrow 2\alpha \cdot (-2\mathbf{S}\pi\mathbf{A} + G_1 + G_2) + (1 - \alpha)\varphi$;
6:     $\pi \leftarrow \text{OT}(\nu, \mu, G, \pi, \epsilon)$ via Algorithm 3;
7:   **end for**

---

## D   Implement Details

A 2-layer graph convolutional network [62] is used as the backbone, and the latent embedding dimension is set as 128. Following [18, 48], the Louvain method [40] is used to divide the initial

---

**Algorithm 3** The main steps of OT via Sinkhorn algorithm

---

**Input:** Node marginal distribution $\nu \in \mathbb{R}^N$, anchor marginal distribution $\mu \in \mathbb{R}^M$, cost matrix $\mathbf{C}^{N \times M}$, initialized transport matrix $\pi$, trade-off parameter $\epsilon$, maximum iteration round $T^{'}$.
**Output:** The transport matrix $\pi \in \mathbb{R}^{N \times M}$.
 1: $K \leftarrow e^{-\mathbf{C}/\epsilon}$;
 2: **for** $t = 1 : T^{'}$ **do**
 3:     $\nu \leftarrow \nu/K\mu$;
 4:     $\mu \leftarrow \mu/K\nu$;
 5: **end for**
 6: $\pi \leftarrow \mathrm{diag}(\nu) K \mathrm{diag}(\mu)$;

---

---

**Algorithm 4** The main steps for low-rank tensor optimization

---

**Input:** Observed tensor $\mathcal{W}$, trade-off parameters $\beta$ and $\mu$, step $\omega$, maximum $\mu_{max} = 10^{10}$, threshold $\epsilon$. Initial the auxiliary variable $\mathcal{G} = \mathbf{0}$, the Lagrange multipliers $\mathcal{H} = \mathbf{0}, \mathcal{Y} = \mathbf{0}$.
**Output:** Consistent matrix $\mathbf{A}$.
 1: **while** not convergent **do**
 2:     Update $\mathcal{G}^{t+1} \leftarrow (\mathcal{L}^t, \mathcal{H}^t)$ by Eq. (20);
 3:     **for** $i = 1 : K$ **do**
 4:       Update $\mathbf{L}_k^{t+1} \leftarrow (\mathbf{A}^t, \mathbf{Z}_k^t, \mathbf{E}_k^t, \mathbf{Y}_k^t, \mathbf{G}_k^t, \mathbf{H}_k^t, \alpha_k^t, \mu^t)$ by Eq. (22);
 5:     **end for**
 6:     Update $\mathbf{A}^{t+1} \leftarrow (\mathbf{L}_k^t, \alpha_k^t)$ by Eq. (24);
 7:     Update $\mathcal{E}^{t+1} \leftarrow (\mathcal{B}^t, \beta/\mu^t)$ by Eq. (26);
 8:     Update $\alpha_k^{t+1} \leftarrow (\mathbf{L}_k^t, \mathbf{A}^t)$ by Eq. (28);
 9:     Check the convergence conditions:
        $\max(||\mathcal{W}^{t+1} - \mathcal{L}^{t+1} - \mathcal{E}^{t+1}||_F^2, ||\mathcal{L}^{t+1} - \mathcal{L}^t||_F^2, ||\mathcal{E}^{t+1} - \mathcal{E}^t||_F^2) \leq \epsilon$
10:     $\triangleright$ When the conditions are satisfied, the iteration ends, otherwise it continues;
11:     $t = t + 1$;
12: **end while**

---

graph to multiple local subgraphs, the number of clients is set as 10. We use Adam as the optimizer with the learning rate set to 0.001. The numbers of communication round and local training rounds are set to 100 and 5, respectively. When the federated training completes, we use the standard linear probing to evaluate the performance of algorithms. The values of ACC and Fscore on test sets are reported. Some trade-off parameters are required to be predefined, $\alpha$ and $\epsilon$ in FGW-OT is fixed as 0.5 and 1. $\lambda$ is tuned in $\{0.1, 5, 10, 50, 70, 100\}$, $\beta$ is varied in $\{0.1, 0.5, 1\}$, the number of anchors is tuned in $\{30, 60, 100, 500, 1000\}$.

# E  The Analysis for the Computational Complexity

The computation complexity mainly originates from two aspects: the FGW-OT for aligning representation spaces and the low-rank tensor optimization for adaptively learning global model parameters. For the FGW-OT, the computation of $G$ in Algorithm 2 takes $\mathcal{O}(N_E M + M^2 N)$, where $N_E$ denotes the number of edges, $M$ denotes the number of anchors, and $N$ denotes the number of nodes. Notably, the adjacency matrices are often sparse, so the computational complexity can be significantly decreased. Likewise, the OT based on Sinkhorn algorithm in Algorithm 3 takes $\mathcal{O}(MN)$. Then, the FGW-OT takes $\mathcal{O}(N_E M + M^2 N + MN)$. For the low-rank tensor optimization, the update of $\mathcal{G}$ with $D_{l,1} \times D_{l,2} \times K$ in Algorithm 4 occupies the dominant computational complexity. First, the FFT and the inverse FFT require $\mathcal{O}(D_{l,1}^2 D_{l,2} \log(K))$, where $D_{l,1}$ and $D_{l,2}$ are the dimensions for the $l$-th layer's parameters, $K$ is the number of clients, and $D_{l,1}$ is the higher dimension. Second, the T-SVD costs $\mathcal{O}(D_{l,1} D_{l,2}^2 K)$. Then, the low-rank tensor optimization takes $\mathcal{O}(D_{l,1}^2 D_{l,2} \log(K) + D_{l,1} D_{l,2}^2 K)$. Overall, the computational complexity for the proposed FedPAM is $\mathcal{O}(N_E M + M^2 N + MN + D_{l,1}^2 D_{l,2} \log(K) + D_{l,1} D_{l,2}^2 K)$.

# F  The Details of Eight Graph Datasets

We conduct the experiments on eight graph datasets, including Cora, CiteSeer, PubMed, Ogbn-Arxiv, Computers, Photo, Physics, Amazon-ratings. Their detailed information about the numbers of nodes, features, edges, classes, the proportions of dataset divisions, and the categories of datasets are described in Table 4. The statistical information of local subgraphs for eight graph datasets is presented in Fig. 7.

Table 4: Descriptions of eight graph datasets.

| Dataset | Nodes | Features | Edges | Classes | Train / Val / Test | Category |
|---|---|---|---|---|---|---|
| Cora | 2,708 | 1,433 | 5,429 | 7 | 20% / 40% / 40% | Citation Network |
| CiteSeer | 3,327 | 3,703 | 4,732 | 6 | 20% / 40% / 40% | Citation Network |
| PubMed | 19,717 | 500 | 44,338 | 3 | 20% / 40% / 40% | Citation Network |
| Ogbn-Arxiv | 169,343 | 128 | 231,559 | 40 | 60% / 20% / 20% | Citation Network |
| Computers | 13,381 | 767 | 245,778 | 10 | 20% / 40% / 40% | Co-purchase Network |
| Photo | 7,487 | 745 | 119,043 | 8 | 20% / 40% / 40% | Co-purchase Network |
| Physics | 34,493 | 8,415 | 247,962 | 5 | 20% / 40% / 40% | Co-author Network |
| Amazon-ratings | 24,492 | 300 | 93,050 | 5 | 50% / 25% / 25% | Rating Network |

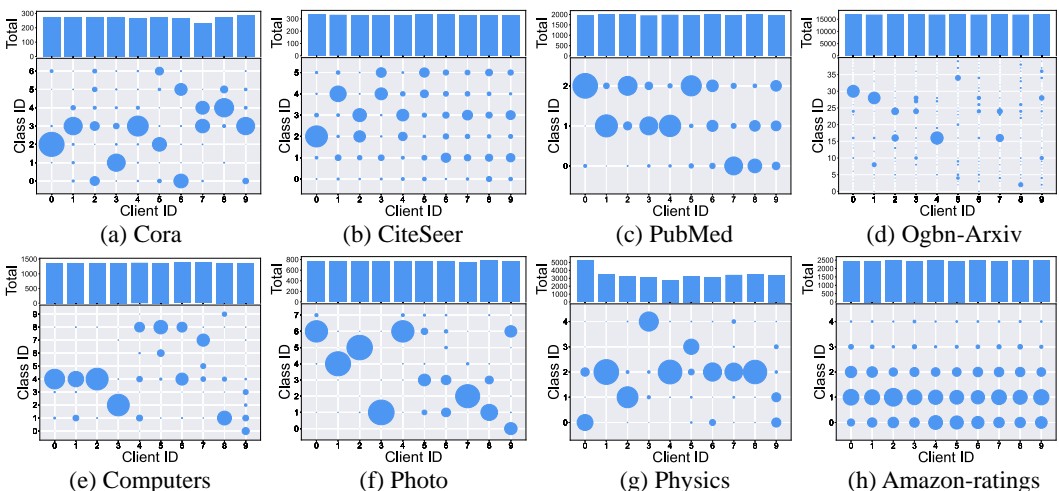

(a) Cora  (b) CiteSeer  (c) PubMed  (d) Ogbn-Arxiv

(e) Computers  (f) Photo  (g) Physics  (h) Amazon-ratings

Figure 7: Statistical information of local subgraphs for eight graph datasets when the number of clients is fixed as 10.

# G  The Performance Comparison Using KNN as the Evaluation Metric

Standard linear probing is sensitive to linearly differentiable features and may not accurately evaluate the model if the data distribution is complex. In addition to using the standard linear probing, we use KNN as the evaluation metric, the results are reported in Table 5. When using Simsiam and BOYL as the SSL losses on CiteSeer dataset, the values of evaluation metrics are much better than that when using standard linear probing. Hence, employing different evaluation methods provide a more comprehensive understanding of models. Fortunately, it can be seen that the proposed FedPAM achieves optimal performance in most cases, proving that embedding output by FedPAM has better discriminative property.

# H  The Performance Comparison with Large-Scale Clients

Performance validation on large-scale clients is an important measure for the scalability of FL methods, we test the performance of various FL methods with 50 and 100 clients on PubMed and Ogbn-Arxiv (OA) datasets, the results are presented in Fig. 8, where SimCLR is used as the SSL loss.

Table 5: Performance comparison on Cora and CiteSeer datasets with three different SSL losses. The KNN is used as evaluation method, where the optimal results are **bolded** and the suboptimal results are underlined.

| SSL | Method | Cora | | CiteSeer | |
|---|---|---|---|---|---|
| | | ACC | Fscore | ACC | Fscore |
| Simsiam | FedAvg | 54.55 | 42.90 | 36.66 | 25.71 |
| | FedProx | 55.17 | 41.80 | 39.59 | 28.01 |
| | MOON | 56.28 | 44.83 | 36.75 | 26.75 |
| | FedU$^2$ | 54.90 | 41.92 | 40.99 | 30.44 |
| | FedPAM | **58.93** | **47.27** | **50.40** | **38.01** |
| SimCLR | FedAvg | 52.49 | 38.05 | 37.77 | 24.60 |
| | FedProx | 53.28 | 39.10 | 36.37 | 22.73 |
| | MOON | 53.01 | 38.74 | 37.68 | 24.80 |
| | FedX | 54.31 | 40.62 | **42.74** | **31.16** |
| | FedU$^2$ | 54.30 | 40.68 | 38.78 | 25.78 |
| | FedPAM | **57.71** | **44.53** | 42.07 | 30.43 |
| BYOL | FedAvg | 64.74 | **53.63** | 44.88 | 34.17 |
| | FedU | 63.35 | 52.59 | 46.25 | 35.33 |
| | FedEMA | 64.20 | 52.53 | 43.30 | 30.62 |
| | Orchestra | 54.12 | 41.25 | 36.89 | 24.65 |
| | FedU$^2$ | 59.87 | 48.18 | 44.58 | 33.33 |
| | FedPAM | **64.99** | 53.08 | **49.25** | **39.68** |

It can be seen that the proposed FedPAM stays ahead of the curve even with a large number of clients, showing that representation space alignment and adaptive global parameter learning are effective. Notably, the superiority of the proposed FedPAM is more obvious in terms of Fscore, indicating that the proposed FedPAM not only has a high overall prediction accuracy, but also shows a better performance on the minority classes.

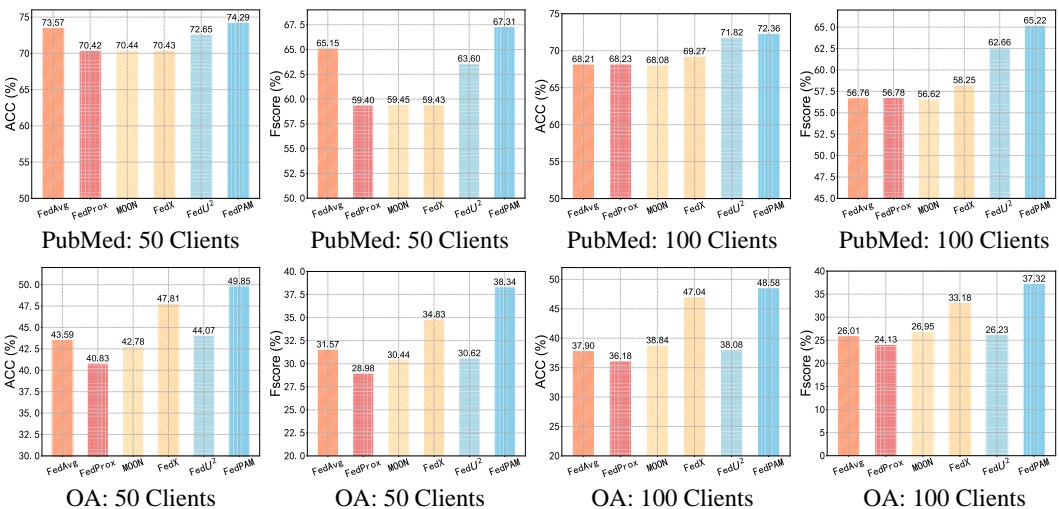

Figure 8: Performance comparison with more clients on PubMed and Ogbn-Arxiv (OA) datasets, where the number of clients is set to 50 and 100, respectively.

# I The Performance Comparison of Plugging the AGPL Module into Existing Works

Notably, the proposed AGPL module is a plug-and-play module, which can be incorporated into existing works and promote their performance. Specifically, we plug AGPL into FedAvg, FedProx,

Table 6: Performance comparison on Cora, CiteSeer, and Ogbn-Arxiv datasets when the AGPL module is plugged into existing works, where the optimal results are **bolded**.

| SSL | Method | Cora | | CiteSeer | | Ogbn-Arxiv | |
| --- | --- | --- | --- | --- | --- | --- | --- |
| | | ACC | Fscore | ACC | Fscore | ACC | Fscore |
| Simsiam | FedAvg | 54.38 | 39.27 | 36.30 | 20.34 | 35.26 | 20.49 |
| | FedAvg+AGPL | **58.56** | **46.67** | **43.29** | **30.60** | **35.33** | **21.21** |
| | FedProx | 55.35 | 40.83 | 36.80 | 21.34 | 34.61 | 20.03 |
| | FedProx+AGPL | **60.15** | **47.07** | **43.75** | **31.58** | **35.26** | **20.49** |
| | MOON | 54.03 | 39.81 | 35.94 | 20.41 | 35.46 | 21.20 |
| | MOON+AGPL | **58.92** | **46.22** | **41.34** | **30.42** | **35.44** | **21.59** |
| SimCLR | FedAvg | 54.65 | 39.95 | 35.48 | 19.62 | 45.58 | 33.08 |
| | FedAvg+AGPL | **56.65** | **44.18** | **41.35** | **27.80** | **47.10** | **37.13** |
| | FedProx | 55.26 | 40.68 | 35.56 | 19.78 | 44.40 | 31.86 |
| | FedProx+AGPL | **56.79** | **46.29** | **41.54** | **28.61** | **47.36** | **35.01** |
| | MOON | 54.56 | 39.81 | 36.30 | 20.34 | 46.23 | 33.96 |
| | MOON+AGPL | **56.49** | **43.74** | **41.82** | **29.26** | **47.91** | **39.04** |
| | FedX | 56.73 | 43.11 | 37.10 | 22.37 | 46.69 | 34.77 |
| | FedX+AGPL | **57.97** | **44.56** | **42.92** | **30.75** | **48.76** | **39.56** |

MOON, and FedX with Simsiam and SimCLR losses, respectively. Table 6 reports the experimental results. It can be observed that the performance of existing works after equipped with AGPL is improved, demonstrating its effectiveness.

