# OpenReview forum: "Unsupervised Federated Graph Learning"
_NeurIPS.cc/2025/Conference — NeurIPS 2025 poster_

### Official Review · Reviewer_z9JY · 2025-06-24

**Clarity:** 3
**Significance:** 3
**Originality:** 3
**Rating:** 5
**Confidence:** 4

**Summary:**

In this paper, the authors proposed a novel method for unsupervised federated graph learning. Specifically, the proposed FedPAM uses a set of anchors to explore the global rep resentation space, then use them to align the local representation spaces. To capture the high-order correlation among multiple clients, the low-rank tensor decomposition is adopted to enhance the consistency between multiple clients. Finally, extensive experiments are conducted to verify the effectiveness for the proposed FedPAM.

**Questions:**

1. The authors should explain why use the low-rank tensor decomposition for obtaining the global model weights.
2. Fig. 4 shows that the proposed FedPAM will degrade when the number of anchors increase. However, according to my understanding, more anchors, more accurate for the global space. Hence, the reason should be elaborated.
3. For an algorithm, the complexity analysis is essential. Hence, the complexity analysis should be provided.

**Ethical Concerns:**

["NO or VERY MINOR ethics concerns only"]

**Final Justification:**

This paper proposed a novel method for unsupervised FGL, the proposed RSA and AGPL modules guaranteed the effective of FedPAM by a large number of experiments. The added generalization and complexity analysis provided a comprehensive understanding for the proposed FedPAM. Likewise, the added experiments further verify the superiority and flexibility for FedPAM. Hence, it's suggested to be accepted.

**Limitations:**

Yes

**Quality:**

3

**Strengths And Weaknesses:**

Strengths:
1. The idea is interesting, using a set of learnable anchors to locate the global space, then adopting them to align local representation spaces, which is appealing.
2. The motivation is well demonstrated by Fig. 1, which verify that the local representation spaces are not aligned and the conventional federated aggregation is ineffective.
3. The paper is well written and the readers can easily follow.

Weaknesses:
1. The motivation of using low-rank tensor decomposition is not clarified.
2. Additionally transfer the anchors can increase the communication burden.
3. The analysis for the experimental results are insufficient.

---

> ### Author Rebuttal · Authors · 2025-07-31
>
> **R1 to Q1:** Thank you for this valuable comment.
> There are two main reasons for using low-rank tensor decomposition. First, most federated aggregation methods perform parameter aggregation at the matrix level and cannot capture high-order correlations between different clients. The so-called high-order correlation refers to the consistency and complementary information across clients, and low-rank tensor decomposition is used to enhance information fusion between clients. Second, the client weight calculation method based on the number of samples often cannot truly reflect the actual degree of client contribution. For example, although some clients have a large number of nodes, they might be extremely heterogeneous and cannot play a positive role in the optimization direction of the global model. Therefore, an adaptive weight learning method is needed.
>
> **R2 to Q2:** Thanks for this insightful comment. We expect to learn a set of representative anchors and reduce the running cost as much as possible, so the number of anchors should be small. Since we do not impose any constraints between anchors, for example, different anchors should be as dissimilar as possible. Therefore, too many anchors may cause the collapse of the anchor space, which is not conducive to the alignment of the representation space across clients.
>
> **R3 to Q3:** Thanks for this meaningful comment. The computation complexity mainly originates from two aspects: the FGW-OT for aligning representation spaces and the low-rank tensor optimization for adaptively learning global model parameters.
> For the FGW-OT, the computation of $G$ (an intermediate variable) takes $\mathcal{O}(N_{E}M+M^{2}N)$, where $N_{E}$ denotes the number of edges, $M$ denotes the number of anchors, and $N$ denotes the number of nodes.
> Notably, the adjacency matrices are often sparse, so the computational complexity can be significantly decreased.
> Likewise, the OT based on Sinkhorn algorithm takes $\mathcal{O}(MN)$.
> Then, the FGW-OT takes $\mathcal{O}(N_{E}M+M^{2}N+MN)$.
> For the low-rank tensor optimization, the update of $\boldsymbol{\mathcal{G}}$ with $D_{l,1}\times D_{l,2}\times K$ occupies the dominant computational complexity.
> First, the FFT and the inverse FFT require $\mathcal{O}(D_{l,1}^{2}D_{l,2}\log(K))$, where $D_{l,1}$ and $D_{l,2}$ are the dimensions for the $l$-th layer's parameters, $K$ is the number of clients, and $D_{l,1}$ is the higher dimension.
> Second, the T-SVD costs $\mathcal{O}(D_{l,1}D_{l,2}^{2}K)$. Then, the low-rank tensor optimization takes $\mathcal{O}(D_{l,1}^{2}D_{l,2}\log(K)+D_{l,1}D_{l,2}^{2}K)$.
> Overall, the computational complexity for the proposed FedPAM is $\mathcal{O}(N_{E}M+M^{2}N+MN+D_{l,1}^{2}D_{l,2}\log(K)+D_{l,1}D_{l,2}^{2}K)$.

---

> > ### Comment · Reviewer_z9JY · 2025-08-03
> >
> > Thanks to the authors' reply, my question has been answered. Furthermore, the other reviewers provided many insightful and interesting comments, which have been basically addressed by the authors, in my opinion. Therefore, I decide to raise the score.

---

> > > ### Author Response · Authors · 2025-08-04
> > > **Response**
> > >
> > > Thanks for your valuable comments and checking our responses. We will further improve the quality in the future!

---

### Official Review · Reviewer_Rhdr · 2025-06-28

**Clarity:** 3
**Significance:** 3
**Originality:** 3
**Rating:** 5
**Confidence:** 4

**Summary:**

This paper addresses unsupervised settings in graph federated learning and defines two main challenges: 1) the misalignment of local representations due to the lack of a common signal (i.e., a supervised label), and 2) the difficulty of addressing heterogeneity across clients when aggregating local parameters on the server. The authors propose a global anchor set to align the representation space across clients, and a low-rank-based weighting mechanism for parameter aggregation. The authors provide a theoretically solid method supported by well-designed experiments.

**Questions:**

See Weaknesses and Questions above.

**Ethical Concerns:**

["NO or VERY MINOR ethics concerns only"]

**Final Justification:**

The rebuttal has addressed my concerns well. My primary concern was the need to demonstrate the isolated effectiveness of the AGPL module, and the authors have successfully shown this in their response, particularly in relation to Table 3. This clarification was crucial. As my concerns are now resolved, I lean towards accepting this paper.

**Limitations:**

yes

**Quality:**

3

**Strengths And Weaknesses:**

**Strengths.**

S1. The paper is well-written and easy to follow, with the authors presenting their ideas clearly.
S2. The paper correctly identifies representation space misalignment as a critical challenge in unsupervised graph federated learning. The proposed solution for this, which leverages a global anchor set, is intuitive and theoretically sound.
S3. The experimental design is well-organized.

**Weaknesses.**

W1. The computational cost of the proposed method is a potential concern, given its reliance on both optimal transport and tensor decomposition, which are known to be computationally intensive operations.
W2. Since $\mathcal{E}$ can be considered as a non-common semantics across other clients' parameter tensor set, it would be more informative if the authors address the potential privacy leak issues from this point.
W3. It would be more helpful to address the potential privacy risks that arise when a client's weight becomes extremely low. Since a low weight signifies that the client is a distinct outlier, this information could potentially be used to infer specific properties about that client.

**Questions.**

Q1. The parameter sensitivity study in Figure 5 shows a strong positive correlation between performance and the hyperparameter $\lambda$, which weights the alignment loss $\mathcal{L}_ {CE}$. This steep trend raises an interesting question about the trade-off between the representation learning from $\mathcal{L}_ {SSL}$ and the representation alignment from $\mathcal{L}_ {CE}$. While $\mathcal{L}_ {SSL}$ is fundamentally necessary to prevent representation collapse in an unsupervised setting, the results suggest its relative contribution can be small as long as a strong alignment signal exists. It would be insightful if the authors could provide further analysis on this trade-off. For example, is there a performance plateau or even a decrease at much higher $\lambda$ values? More importantly, what is the minimum relative weight of the SSL loss required to maintain stable and non-trivial representations?

Q2. The AGPL module is presented as a novel method for adaptive parameter aggregation that tackles the shortcomings of standard federated averaging. To better isolate and demonstrate its effectiveness as a generalizable component, it would be highly compelling to see an experiment where AGPL is applied as an "add-on" to other baseline methods, such as FedAvg. Such an experiment would provide clearer evidence of AGPL's standalone contribution, independent of the RSA module, and strengthen the paper's claims about its utility.

---

> ### Author Rebuttal · Authors · 2025-07-31
>
> **R1 to W1:** Thank you for this good comment. The computation complexity mainly originates from two aspects: the FGW-OT for aligning representation spaces and the low-rank tensor optimization for adaptively learning global model parameters.
> For the FGW-OT, the computation of $G$ (an intermediate variable) takes $\mathcal{O}(N_{E}M+M^{2}N)$, where $N_{E}$ denotes the number of edges, $M$ denotes the number of anchors, and $N$ denotes the number of nodes.
> Notably, the adjacency matrices are often sparse, so the computational complexity can be significantly decreased.
> Likewise, the OT based on Sinkhorn algorithm takes $\mathcal{O}(MN)$.
> Then, the FGW-OT takes $\mathcal{O}(N_{E}M+M^{2}N+MN)$.
> For the low-rank tensor optimization, the update of $\boldsymbol{\mathcal{G}}$ with $D_{l,1}\times D_{l,2}\times K$ occupies the dominant computational complexity.
> First, the FFT and the inverse FFT require $\mathcal{O}(D_{l,1}^{2}D_{l,2}\log(K))$, where $D_{l,1}$ and $D_{l,2}$ are the dimensions for the $l$-th layer's parameters, $K$ is the number of clients, and $D_{l,1}$ is the higher dimension.
> Second, the T-SVD costs $\mathcal{O}(D_{l,1}D_{l,2}^{2}K)$. Then, the low-rank tensor optimization takes $\mathcal{O}(D_{l,1}^{2}D_{l,2}\log(K)+D_{l,1}D_{l,2}^{2}K)$.
> Overall, the computational complexity for the proposed FedPAM is $\mathcal{O}(N_{E}M+M^{2}N+MN+D_{l,1}^{2}D_{l,2}\log(K)+D_{l,1}D_{l,2}^{2}K)$.
>
> In addition, we also report the running time on Cora, CiteSeer, and PubMed in Table 1. We can see that the proposed FedPAM does indeed have lower runtime efficiency than most comparison algorithms, but in order to achieve satisfactory results, a longer runtime is necessary. At the same time, it is worth noting that compared to FedU$^2$, FedPAM is still acceptable in terms of runtime.
>
> *Table 1: Comparison of running time (seconds) on three datasets.*
> | SSL      | Method    | Cora    | CiteSeer | PubMed   |
> |----------|-----------|---------|----------|----------|
> | **Simsiam** | FedAvg    | 47.72   | 49.23    | 48.32    |
> |          | FedProx   | 58.98   | 59.72    | 60.50    |
> |          | MOON      | 67.28   | 67.39    | 66.84    |
> |          | FedU$^2$  | 2040.83 | 2041.00  | 2086.69  |
> |          | FedPAM    | 710.12  | 711.36   | 712.73   |
> | **SimCLR**  | FedAvg    | 41.77   | 42.00    | 41.43    |
> |          | FedProx   | 50.16   | 50.18    | 50.26    |
> |          | MOON      | 66.03   | 66.41    | 67.23    |
> |          | FedX      | 103.54  | 92.16    | 105.75   |
> |          | FedU$^2$  | 2092.78 | 2095.67  | 2102.21  |
> |          | FedPAM    | 582.62  | 602.36   | 605.89   |
> | **BYOL**    | FedAvg    | 62.14   | 62.02    | 62.37    |
> |          | FedU      | 62.10   | 63.78    | 62.70    |
> |          | FedEMA    | 63.33   | 62.45    | 63.00    |
> |          | Orchestra | 90.61   | 86.62    | 155.13   |
> |          | FedU$^2$  | 2137.52 | 2136.59  | 2207.17  |
> |          | FedPAM    | 828.67  | 826.81   | 830.01   |
>
> **R2 to W2:** Thank you for this interesting suggestion.
> Privacy protection is an important issue in federated learning, where $\mathcal{E}$ contains client-specific information. Once the centralized server is attacked, this may cause client privacy leakage. The current tensor decomposition method has an explicit bias tensor $\mathcal{E}$. Therefore, in order to protect the client's privacy information from being exposed, we can use the tensor Tucker decomposition to decompose a core (low-rank) tensor and a set of factor (bias) matrix. The mathematic form is written as
> >$\mathcal{W}=\mathcal{L} \times_1 U_1 \times_2 U_2 \cdots \times_M U_M$,
>
> where $\mathcal{W}$, $\mathcal{L}$, and ${U_{i}}\_{i=1}^{M}$ denote the original tensor, the low-rank tensor, and a set of factor matrix. Since the factor matrices integrate the information of multiple clients, they will not expose client-specific information.
> Limited by the rebuttal time, we will further investigate a novel method for unsupervised FGL with strict privacy protection in the future.
>
> **R3 to W3:** Thanks for this insightful comment.
> In federated learning, low client weights can lead to privacy leaks (for example, by inferring client data distribution or sample information through reverse engineering). To avoid this problem, low-weight clients can be clustered with similar clients based on their data distribution or model parameter similarity to form virtual "super clients." Parameters are first aggregated within the group, and then the group participates in global aggregation, masking individual contributions. Clients within the group share the aggregated weights, preventing individual clients from having too low a weight. This is a very interesting research topic that we will further explore in future work.
>
> **R1 to Q1:** Thank you for this interesting question.
> We continue to increase the value of $\lambda$ to test whether FedPAM will encounter a performance bottleneck. The experimental results are reported in Table 2. We can see that when $\lambda$ rises to 500 or 600, FedPAM encounters a performance bottleneck. However, it is worth noting that a larger value of $\lambda$ is conducive to achieving satisfactory performance, thanks to the global anchor-guided representation space alignment. From the experimental results, to achieve satisfactory results, the weight of $\lambda$ relative to SSL weight should be at least 10.
>
> *Table 2: Performance comparison with different values of $r$.*
> | Dataset   | $\lambda=100$ | $\lambda=200$ | $\lambda=300$ | $\lambda=400$ | $\lambda=500$ | $\lambda=600$ | $\lambda=700$ |
> |-----------|--------------:|--------------:|--------------:|--------------:|--------------:|--------------:|--------------:|
> | Cora      |         62.83 |         63.29 |         65.92 |         64.96 |         67.65 |         67.56 |         37.10 |
> | CiteSeer  |         43.90 |         46.59 |         50.64 |         52.65 |         53.24 |         36.30 |         36.52 |
>
> **R2 to Q2:** Thank you for this good advice. In essence, the proposed AGPL module is a plug-and-play component. To verify its effectiveness, we stitch the AGPL module with the existing federated learning algorithm.
> Notably, we focus on testing the federated learning algorithm without special aggregation methods on the server side.
> The experimental results are reported in Table 3.
> We can see that when the existing federated learning algorithm is combined with the proposed AGPL module, the performance will be improved, which illustrates the effectiveness of the AGPL module.
>
> *Table 3: Performance comparison when the proposed AGPL is combined with different FL methods.*
> | SSL      | Method         | Cora ACC | Cora Fscore | CiteSeer ACC | CiteSeer Fscore | Ogbn-Arxiv ACC | Ogbn-Arxiv Fscore |
> |----------|----------------|----------|-------------|--------------|-----------------|----------------|-------------------|
> | **Simsiam** | FedAvg+AGPL   | 57.38    | 47.21       | 42.27        | 30.42           | 35.58          | 21.32             |
> |          | FedProx+AGPL   | 58.25    | 48.02       | 42.75        | 30.68           | 35.22          | 21.01             |
> |          | MOON+AGPL      | 56.70    | 46.56       | 41.02        | 28.83           | 35.58          | 21.33             |
> |          | FedPAM         | **61.40**    | **52.85**       | **49.53**        | **38.73**           | **35.97**          | **23.07**             |
> | **SimCLR**  | FedAvg+AGPL   | 56.65    | 44.18       | 41.35        | 27.80           | 47.65          | 35.45             |
> |          | FedProx+AGPL   | 56.86    | 44.43       | 41.58        | 27.93           | 46.97          | 35.86             |
> |          | MOON+AGPL      | 55.81    | 43.83       | 42.10        | 28.12           | 47.80          | 36.67             |
> |          | FedX+AGPL      | 57.53    | 44.96       | 42.86        | 30.23           | 48.01          | 36.95             |
> |          | FedPAM         | **58.62**    | **45.21**       | **43.90**       | **31.61**           |**48.19**          | **37.25**            |

---

> > ### Comment · Reviewer_Rhdr · 2025-08-04
> >
> > Thank you for the rebuttal, which effectively addresses my concerns, particularly regarding Table 3. I hope this clarification will be incorporated into the final version of the paper. I have raised my score.

---

> > > ### Author Response · Authors · 2025-08-04
> > > **Response**
> > >
> > > Many thanks for provding a lot of insightful suggestions and checking our responses. Thank you for your recognition of our work, which is the greatest motivation for us to move forward! We will add Table 3 in the revised version.

---

### Official Review · Reviewer_pPE4 · 2025-06-30

**Clarity:** 2
**Significance:** 3
**Originality:** 2
**Rating:** 4
**Confidence:** 3

**Summary:**

This paper focuses on unsupervised federated graph learning. The authors propose a tailored framework named FedPAM consisting of two components: Representation Space Alignment (RSA) and Adaptive Global Parameter Learning (AGPL). Experiments on eight graph datasets to demonstrate the superiority of the proposed model compared with other baselines.

**Questions:**

Q1: What is the meaning of $\mathbf{L}_k$ in Eq. (17)?

Q2: Could the authors explain the intuition of Eq. (17)?

Q3: How do we compute $\mathcal{E}$?

Q4: What is the value of $r$ in the experiments? How does it affect the performance of the proposed method?

Q5: In Table 1 and Table 2, only some baselines are included. Could the authors explain why?

**Ethical Concerns:**

["NO or VERY MINOR ethics concerns only"]

**Final Justification:**

After the rebuttal, my concerns are addressed by the authors' response. The authors are expected to modify the submission accordingly in the revised version of their paper.

**Limitations:**

Yes

**Quality:**

2

**Strengths And Weaknesses:**

***Strengths***

S1: The studied problem is important and promising.

S2: The authors conduct experiments on eight graph datasets to evaluate the performance of the proposed method.

***Weaknesses***

W1: GNN formulation. The computation of a GNN in Eq. (1) looks questionable. The authors may check how previous studies formulate the forward pass of GNNs.

W2: Using the product $\mathbf{P}_k \mathbf{Z}^T_k$ to measure the similarity between the anchors and nodes seems problematic. The product will depend on $\mathbf{P}_k$’s and $\mathbf{Z}_k$'s norms as well. A better metric is cosine similarities.

W3: Notations. A letter should not denote different variables in the paper. For example, $\lambda$ in Eq. (8) and Eq. (16); $\mathbf{A}$ to denote the adjacency matrix, the relationships between anchors, and the optimal global GNN parameters.

W4: Self-supervised learning strategies. The authors adopt Simsiam, SimCLR, and BYOL. It is quite weird not to use graph self-supervised learning methods.

---

> ### Author Rebuttal · Authors · 2025-07-31
>
> **R1 to W1:** Thank you for this rigorous advice. We found that the current form of writing is somewhat unclear. After checking previous work, we changed the forward propagation formula to the following form:
> >$\mathbf{z}_i^{l+1} =\sigma(\mathbf{z}_i^l, AGG(\mathbf{z}\_{j}^l; j\in\mathcal{N}_i)), \forall l\in[L]$,
>
> where $\sigma(\cdot)$ denotes the activation function, $A G G(\cdot)$ denotes the aggregation method, $\mathcal{N}_{i}$ denotes  the neighborhood node set for the $i$-th node.
>
> **R2 to W2:** Thanks for this good question. Using cosine similarity is a more common method. In Eq. (11), we also use cosine similarity to calculate the similarity between $\mathbf{P\}_{k}$ and $\mathbf{Z}\_{k}$, and then compare the experimental results of different calculation methods in Table 1.
> The difference in results between using the product and using cosine similarity is not significant. This is because entropy regularization smooths the solution space in the OT calculation, which has a certain normalization effect. Therefore, using the product calculation can still guarantee satisfactory results.
>
> *Table 1: Performance conparison with different similarity calculation methods.*
> | SSL       | Method          | Cora (ACC) | Cora (Fscore) | CiteSeer (ACC) | CiteSeer (Fscore) |
> |-----------|-----------------|----------|-------------|--------------|-----------------|
> | **Simsiam**  | FedPAM-Product  | 61.40    | 52.85       | 49.53        | 38.73           |
> |           | FedPAM-Cosine   | 61.85    | 53.12       | 49.89        | 39.31           |
> | **SimCLR**   | FedPAM-Product | 58.62    | 45.21       | 43.90        | 31.61           |
> |           | FedPAM-Cosine   | 58.93    | 45.83       | 43.90        | 31.61           |
>
> **R3 to W3:** Thanks for this rigorous comment. We will further improve the notations in the following revisions.
>
> **R4 to W4:** Thank you for this good comment. It is worth noting that Simsiam, SimCLR, and BYOL are the three most widely used self-supervised learning methods, applicable in almost all fields. In the field of graphs, the augmented view is achieved by perturbing the edges and masking some feature elements, while the contrast loss is no different, such as work [1] and work [2]. Of course, there are some more complex graph self-supervised learning methods, however, our focus is on how to achieve unsupervised joint modeling in distributed graph scenarios, and local contrast loss is not the focus of research.
>
> [1] Deep Graph Contrastive Representation Learning
>
> [2] Deep Graph Infomax
>
> **R1 to Q1:** Thanks for this rigorous comment. $\mathbf{L}_{k}$ is the $k$-th slice of low-rank tensor $\mathcal{L}$ corresponding to the parameters of the kth client.
>
> **R2 to Q2:** Thanks for this good comment. The original intention behind the optimization objective of Eq. (17) comes from two aspects:
> The original intention behind the optimization objective of Eq. (17) stems from two aspects: First, matrix-based aggregation methods cannot capture higher-order correlations between different clients, and the complementary information between clients cannot be fully utilized. Therefore, we stack the parameters of different clients into a third-order tensor and use low-rank tensor decomposition in high-dimensional space to explore the complementarity between clients.
> Second, traditional federated aggregation performs weighted aggregation based on sample size, which cannot adaptively learn optimal parameters. Therefore, we aim to learn a set of adaptive client weights while adaptively aggregating model parameters from different clients to obtain optimal global model parameters. Based on these two aspects, we establish a joint optimization loss, as shown in Eq. (17).
>
> **R3 to Q3:** Thanks for the good question. When $\mathcal{E}$ is updated, the other variables are viewed as constants, the subproblem is formulated as
> $$
> \min_{\boldsymbol{\mathcal{E}}}\beta||\boldsymbol{\mathcal{E}}||_{1} + \langle\boldsymbol{\mathcal{Y}},\boldsymbol{\mathcal{W}}-\boldsymbol{\mathcal{L}}-\boldsymbol{\mathcal{E}}\rangle + \frac{\mu}{2}||\boldsymbol{\mathcal{W}}-\boldsymbol{\mathcal{L}}-\boldsymbol{\mathcal{E}}||\_{F}^{2}
> $$
>
> $$
> = \min\_{\boldsymbol{\mathcal{E}}}\beta||\boldsymbol{\mathcal{E}}||_{1} + \frac{\mu}{2}||\boldsymbol{\mathcal{E}}-(\boldsymbol{\mathcal{W}}-\boldsymbol{\mathcal{L}}+\boldsymbol{\mathcal{Y}}/\mu)||\_{F}^{2}
> $$
>
> For the $l_1$-norm subproblem, we adopt the solution proposed in the work [3], then obtain the update rule for $\mathcal{E}$, the details can be referred in [3].
>
> [3] The Augmented Lagrange Multiplier Method for Exact Recovery of Corrupted Low-Rank Matrices
>
> **R4 to Q4:** Thanks for this valuable comment. $r$ is a constant to smooth the distribution of client weights, the larger the value of $r$, the smoother the client weights, otherwise the sharper they are. it is set to 2 on all datasets.
> We test the FedPAM's performance with differen $r$ in Table 2, it can be seen that most of the experimental results are best when $r=2$, which shows that a slightly smooth client weight distribution is conducive to achieving considerable results. A smooth client weight distribution is conducive to balancing the information of multiple clients, while a sharp client weight distribution will amplify the influence of some clients, resulting in biased fusion.
>
> *Table 2: Performance compasiron (ACC \%) with different values of $r$.*
> | SSL     | Dataset  | r=0.5 | r=1     | r=2     | r=5  | r=10 |
> |---------|----------|-------|---------|---------|------|------|
> | **Simsiam** | Cora     | 58.82 | **61.56** | 61.40  | 60.21| 57.89|
> |         | CiteSeer | 46.91 | 48.52   | **49.53**| 47.21| 47.10|
> | **SimCLR**  | Cora     | 57.92 | 58.35   | **58.62**| 58.02| 58.02|
> |         | CiteSeer | 41.92 | **44.05** | 43.90  | 42.12| 41.56|
> | **BYOL**    | Cora     | 64.92 | 65.56   | **66.01**| 65.21| 64.23|
> |         | CiteSeer | 47.10 | 48.32   | **48.81**| 47.95| 46.92|
>
> **R5 to Q5:** Thanks for this valuable comment.
> First, unsupervised federated learning is still an unexplored field, with limited works and few available comparison methods. Second, we refer to the baseline setting method of unsupervised federated learning work [5] and conduct three types of comparisons based on model architecture. When using different SSL losses, the comparison methods used are also different, which is determined by the model design. For example, FedU, FedEMA, and Orchestra are based on the architecture with online network and target network.
> Therefore, when the SSL loss is Simsiam or SimCLR, these methods are not applicable.
> MOON uses the output of the global model to calibrate the output of the local model, and is also not applicable to the online network and target network architecture.
> In summary, when using different SSL losses, only some baselines are available.
>
> [5] Rethinking the representation in federated unsupervised learning with non-iid data

---

> > ### Comment · Reviewer_pPE4 · 2025-08-06
> >
> > Thanks for your response. The new experimental results are helpful. I am willing to raise my score to borderline accept. However, I still have the following unsolved concerns.
> >
> > - W1: It seems that there should be a learnable layer before the activation function in Equation (1); otherwise, the representations at each layer of a GNN will have to be of the same dimension.
> >
> > - W2: Thanks for the new results. Double-check if the values in Table 1 are not mistaken.
> >
> > - W4: I still think graph self-supervised learning methods should be used in the experiments since they can be seamlessly adopted on graph data. I understand running experiments are onerous during the intensive rebuttal period, so I encourage the authors to run them in the future.
> >
> > BTW, the authors mentioned several times that my comments are rigorous. I am not sure if I have an overhigh expectation for a NeurIPS submission. Everytime I start to read a paper, I always assume that the authors are trying to make it readable. I think correct preliminaries, proper notations, and clear definitions are the basic requirements for a research paper. They encourage readers to read and follow your methodology.
> >
> > Best,
> > Reviewer pPE4

---

### Official Review · Reviewer_9QFW · 2025-07-02

**Clarity:** 3
**Significance:** 2
**Originality:** 2
**Rating:** 3
**Confidence:** 3

**Summary:**

This paper proposes a new framework, FedPAM, to tackle unsupervised federated graph learning (FGL). The authors identify two major challenges in this setting: (1) the difficulty in aligning the representation spaces across clients due to lack of shared semantics, and (2) the limitations of conventional averaging in aggregating heterogeneous local models. To address these, FedPAM integrates two core components: (1) Representation Space Alignment (RSA): Uses learnable global anchor points and fused Gromov-Wasserstein Optimal Transport to align local client embeddings into a shared global space. (2) Adaptive Global Parameter Learning (AGPL): Stacks local model parameters as tensors and applies low-rank tensor optimization to adaptively fuse them, capturing higher-order correlations among clients. The framework is evaluated on eight graph datasets with three types of self-supervised losses. The results show consistent performance improvements over existing federated and unsupervised baselines.

**Questions:**

Please see Weaknesses

**Ethical Concerns:**

["NO or VERY MINOR ethics concerns only"]

**Final Justification:**

The additional experiments and clarifications helped address some of my concerns, particularly regarding scalability and the broader applicability of the proposed components. While I still find the motivation somewhat generic to the unsupervised FGL setting.

**Limitations:**

Yes

**Quality:**

2

**Strengths And Weaknesses:**

Strengths:
- The paper focuses on a novel unsupervised federated graph learning setting, which is a real need for privacy-preserving, scalable graph learning.

- Experiments are conducted on eight diverse graph datasets, using three SSL losses, and compared against a wide range of baselines. Ablation studies, parameter sensitivity, and visualizations are included to support the method’s effectiveness and robustness.

- The paper is overall easy to follow and well-organized.

Weaknesses:
- Both FGW-OT and tensor decomposition with ADMM are computationally expensive. This limits the implementation of FedPAM to large-scale FGL systems.

- The proposed challenges of “Inability to align the representation spaces of various clients” and “Unable to adaptively learn the optimal parameters of global model” fail to capture the unique characteristics of the unsupervised FGL problem. The experimental settings also follow the normal graph partitioning strategies in previous supervised FGL problems. It would be helpful if the authors could validate the challenges in the experiment section.

- It seems that both the proposed alignment and aggregation techniques are independent with the unsupervised learning nature of the problem. Can we use them in normal supervised FGL systems?

- It would be helpful to add experimental results of the complexity or time cost of FedPAM.

- The results are reported as point estimates without error bars or statistical testing, which limits confidence in the observed gains.

---

> ### Author Rebuttal · Authors · 2025-07-31
>
> **R1 to W1:** Thank you for this valuable comment. We test the proposed FedPAM with more clients. The experimental results are reported in Tables 1 and 2.
> We can see that FedPAM can still maintain its leading position in large-scale client scenarios. Although FGW-OT and low-rank tensor decomposition bring a large amount of computation, it is worth it.
>
> *Table 1: Performance comparison with 50 clients, where the SimCLR is employed in local training.*
> | Method   | PubMed (ACC) | PubMed (Fscore) | OA (ACC) | OA (Fscore) |
> |----------|------------|---------------|--------|-----------|
> | FedAvg  |73.57 | 65.15| 43.59  | 31.57 |
> | FedProx | 70.42| 59.40| 40.83  | 28.98 |
> | MOON | 70.44 | 59.45 | 42.78  | 30.44 |
> | FedX |70.43 | 59.43 | 47.81  | 34.83|
> | FedU$^2$ | 72.65| 63.60| 44.07  | 30.62     |
> | **FedPAM** | **74.29** | **67.31**| **49.85** | **38.34** |
>
> *Table 2: Performance comparison with 100 clients, where the SimCLR is employed in local training.*
> | Method   | PubMed (ACC) | PubMed (Fscore) | OA (ACC) | OA (Fscore) |
> |----------|------------|---------------|--------|-----------|
> |FedAvg| 68.21| 56.76| 37.90 |26.01|
> |FedProx| 68.23| 56.78| 36.18 |24.13|
> |MOON| 68.08| 56.62| 38.84 |26.95|
> |FedX| 69.27| 58.25|47.04 |33.18|
> | FedU$^2$ | 71.82| 62.66|38.08| 26.23|
> |**FedPAM**| **72.36**| **65.22** |**48.58**| **37.32**|
>
> **R2 to W2:** Thank you for this insightful comment.
> First, due to the problem of data heterogeneity, the problem of representation space misalignment exists in both supervised and unsupervised scenarios. In the unsupervised scenario, this problem is more prominent. We centrally train a GNN model on the Cora dataset to output a representation space close to the ideal. Then we train a global model through FedAvg in the supervised and unsupervised scenarios respectively. Through wasserstein distance (WD) calculation, we found that the WD between the global representation output by the supervised FedAvg and the centrally trained representation is 3.67, while the WD between the global representation output by the unsupervised FedAbg and the centrally trained representation is 9.05. Therefore, in the unsupervised scenario, the representation space misalignment is much worse. At the same time, we also illustrate the problem of representation space misalignment in the unsupervised scenario in Figure 1(a).
> Second, the inability to adaptively learn optimal global model parameters is a problem common in both supervised and unsupervised federated graph learning. The proposed AGPL is a plug-and-play component that can also be used in supervised federated graph scenarios. In the response to weakness 3, we conducted experiments demonstrating its effectiveness in supervised scenarios. In summary, the proposed FedPAM addresses both the specific challenges of unsupervised federated graphs and the general challenges of federated graph learning, making it a significant work.
>
> **R3 to W3:** Thanks for this interesting question.
> In fact, both of the proposed Representation Space Alignment (RSA) and Adaptive Global Parameter Learning (AGPL) can be used in supervised FGL systems. We compared the supervised federated graph learning algorithms with the proposed FedPAM and the experimental results are reported in Table 3.
> In the local training, the supervised CE loss is added into training loss.
> Notably, RSA is used to align the representation spaces of different clients, which is especially important in unsupervised scenarios. In supervised scenarios, since the label semantics of different clients are shared, this in itself is a strong alignment signal.
> Therefore, when only the RSA module is used (FedPAM-RSA), the effect is only slightly improved.
> AGPL aims to explore high-level correlations between clients and adaptively learn optimal global parameters, which is applicable in both unsupervised and supervised scenarios. Therefore, its performance is superior to FedPAM-RSA. When using the complete FedPAM, FedPAM achieves remarkable results, even surpassing SOTA unsupervised FGL on the CiteSeer dataset.
>
> *Table: 3 Performance comparison with various supervised FGL methods.*
> | Method | Cora (ACC) | Cora (Fscore) | CiteSeer (ACC) | CiteSeer (Fscore) | PubMed (ACC) | PubMed (Fscore) |
> |---------------|----------|-------------|--------------|-----------------|------------|---------------|
> |FedAvg|73.59|72.04| 68.85 |68.09|82.42|80.20|
> |FedProx| 74.29    | 72.89 | 69.65 | 68.75 | 82.43| 80.21|
> |MOON| 74.22    | 71.88| 68.78 | 68.18 | 82.49| 80.41|
> |FGSSL| 74.47    | 72.94| 70.02 | 69.19 | 82.38| 79.88 |
> |FedPUB| 75.35    | 73.08| 65.36| 62.66| 82.67 | 80.43  |
> |FedTAD| 74.29    | 71.57 | $\underline{71.32}$ | $\underline{70.43}$    | 82.72      | 80.26         |
> |FedATH| **77.90**| **76.93**   | 70.32  | 69.25  | **84.06**  | **82.71**     |
> |FedPAM-RSA|74.10    | 72.69       | 69.22 | 68.37           | 82.56      | 80.32         |
> |FedPAM-AGPL| $\underline{75.83}$ | $\underline{73.69}$ | 71.22    | 70.10           | 83.03      | 81.21         |
> |FedPAM|75.52|73.40|**71.63** |**70.52** |$\underline{83.56}$  | $\underline{81.32}$ |
>
>
> **R4 to W4:** Thank you for this valuable comment. The computation complexity mainly originates from two aspects: the FGW-OT for aligning representation spaces and the low-rank tensor optimization for adaptively learning global model parameters.
> For the FGW-OT, the computation of $G$ (an intermediate variable) takes $\mathcal{O}(N_{E}M+M^{2}N)$, where $N_{E}$ denotes the number of edges, $M$ denotes the number of anchors, and $N$ denotes the number of nodes.
> Notably, the adjacency matrices are often sparse, so the computational complexity can be significantly decreased.
> Likewise, the OT based on Sinkhorn algorithm takes $\mathcal{O}(MN)$.
> Then, the FGW-OT takes $\mathcal{O}(N_{E}M+M^{2}N+MN)$.
> For the low-rank tensor optimization, the update of $\boldsymbol{\mathcal{G}}$ with $D_{l,1}\times D_{l,2}\times K$ occupies the dominant computational complexity.
> First, the FFT and the inverse FFT require $\mathcal{O}(D_{l,1}^{2}D_{l,2}\log(K))$, where $D_{l,1}$ and $D_{l,2}$ are the dimensions for the $l$-th layer's parameters, $K$ is the number of clients, and $D_{l,1}$ is the higher dimension.
> Second, the T-SVD costs $\mathcal{O}(D_{l,1}D_{l,2}^{2}K)$. Then, the low-rank tensor optimization takes $\mathcal{O}(D_{l,1}^{2}D_{l,2}\log(K)+D_{l,1}D_{l,2}^{2}K)$.
> Overall, the computational complexity for the proposed FedPAM is $\mathcal{O}(N_{E}M+M^{2}N+MN+D_{l,1}^{2}D_{l,2}\log(K)+D_{l,1}D_{l,2}^{2}K)$.
>
> Furthermore, we report the running time of compared methods on three datasets in Table 4. We can see that the proposed FedPAM does indeed have lower runtime efficiency than most comparison algorithms, but in order to achieve satisfactory results, a longer runtime is necessary. At the same time, it is worth noting that compared to FedU$^2$, FedPAM is still acceptable in terms of runtime.
>
> *Table 4: Comparison of running time (seconds) on three datasets.*
> | SSL      | Method    | Cora    | CiteSeer | PubMed   |
> |----------|-----------|---------|----------|----------|
> | **Simsiam** | FedAvg    | 47.72   | 49.23 | 48.32    |
> |          | FedProx   | 58.98   | 59.72    | 60.50|
> |          | MOON      | 67.28   | 67.39    | 66.84 |
> |          | FedU$^2$  | 2040.83 | 2041.00  | 2086.69  |
> |          | FedPAM    | 710.12  | 711.36   | 712.73   |
> | **SimCLR**  | FedAvg    | 41.77   | 42.00    | 41.43    |
> |          | FedProx   | 50.16   | 50.18    | 50.26    |
> |          | MOON      | 66.03   | 66.41    | 67.23    |
> |          | FedX      | 103.54  | 92.16    | 105.75   |
> |          | FedU$^2$  | 2092.78 | 2095.67  | 2102.21  |
> |          | FedPAM    | 582.62  | 602.36   | 605.89   |
> | **BYOL**    | FedAvg    | 62.14   | 62.02    | 62.37    |
> |          | FedU      | 62.10   | 63.78    | 62.70    |
> |          | FedEMA    | 63.33   | 62.45    | 63.00    |
> |          | Orchestra | 90.61   | 86.62    | 155.13   |
> |          | FedU$^2$  | 2137.52 | 2136.59  | 2207.17  |
> |          | FedPAM    | 828.67  | 826.81 | 830.01   |
>
> **R5 to W5:** Thanks for this good comment. In fact, we repeated each experiment five times and reported the mean values. To save space, we did not report the variances. Here, we supplemented the variances for the Cora in Table 5.
> It can be seen that the variance of the FGL algorithm is relatively small, and these algorithms are generally stable.
>
> *Table 5: Performance comparison with standard deviations*
> | SSL       | Method          | Cora ACC          | Cora Fscore       |
> |-----------|-----------------|-------------------|-------------------|
> | **Simsiam** | FedAvg          | 54.38$\pm$0.45    | 39.27$\pm$0.37    |
> |           | FedProx         | 55.35$\pm$0.42    | 40.83$\pm$0.47    |
> |           | MOON            | 54.03$\pm$0.68    | 39.81$\pm$0.81    |
> |           | Fed$\operatorname{U}^2$ | $\underline{56.04\pm0.82}$ | $\underline{42.65\pm0.79}$ |
> |           | FedPAM          | **61.40$\pm$0.67** | **52.85$\pm$0.61** |
> | **SimCLR**  | FedAvg          | 54.65$\pm$0.51    | 39.95$\pm$0.45    |
> |           | FedProx         | 55.26$\pm$0.67    | 40.68$\pm$0.66    |
> |           | MOON            | 54.56$\pm$0.44    | 39.81$\pm$0.47    |
> |           | FedX            | 56.73$\pm$0.26    | 43.11$\pm$0.28    |
> |           | Fed$\operatorname{U}^2$ | $\underline{57.09\pm0.57}$ | $\underline{43.45\pm0.50}$ |
> |           | FedPAM          | **58.62$\pm$0.66** | **45.21$\pm$0.57** |
> | **BYOL**    | FedAvg          | 63.01$\pm$0.77    | 50.51$\pm$0.69    |
> |           | FedU            | 64.06$\pm$0.81    | 51.92$\pm$0.77    |
> |           | FedEMA          | 63.53$\pm$0.68    | 51.23$\pm$0.65    |
> |           | Orchestra       | 54.47$\pm$1.11    | 39.63$\pm$0.98    |
> |           | Fed$\operatorname{U}^2$ | $\underline{65.11\pm0.62}$ | $\underline{54.16\pm0.58}$ |
> |           | FedPAM          | **66.01$\pm$0.87** | **55.23$\pm$0.85** |

---

> > ### Comment · Area_Chair_5rrS · 2025-08-06
> >
> > Dear Reviewer,
> >
> > Thank you for reviewing the paper. Please respond to the authors' rebuttal and indicate whether your concerns have been addressed or not, rather than simply pressing the acknowledgement button. Your explicit feedback is important for improving the quality of the paper.
> >
> > Best regards,
> >
> > AC

---

> > ### Comment · Reviewer_9QFW · 2025-08-08
> >
> > Thank you for the detailed and thoughtful rebuttal. The additional experiments and clarifications helped address some of my concerns, particularly regarding scalability and the broader applicability of the proposed components.

---

> > > ### Author Response · Authors · 2025-08-09
> > > **Responses**
> > >
> > > Many thanks again for your valuable comments and recognition for our rebuttal, which greatly encourage us to move forward.

---

### Official Review · Reviewer_cZbY · 2025-07-03

**Clarity:** 3
**Significance:** 2
**Originality:** 3
**Rating:** 4
**Confidence:** 4

**Summary:**

This paper addresses the unsupervised Federated Graph Learning (FGL) problem and proposes a novel framework named FedPAM, which consists of two key modules: (1) Representation Space Alignment (RSA), which aligns the representation spaces of local clients using learnable anchors and Fused Gromov-Wasserstein Optimal Transport (FGW-OT); and (2) Adaptive Global Parameter Learning (AGPL), which fuses local model parameters in a low-rank tensor space via tensor decomposition. The authors validate the method through extensive experiments on eight graph datasets using various self-supervised learning (SSL) strategies.

**Questions:**

See weaknesses.

**Ethical Concerns:**

["NO or VERY MINOR ethics concerns only"]

**Final Justification:**

In the rebuttal period, the authors made good clarification and improved this paper, such as the theoretical analysis. After consideration, I give a boardline acc to this work.

**Limitations:**

The author should include the limitation section to discuss limitations and potential negative societal impact of this work.

**Quality:**

3

**Strengths And Weaknesses:**

Strengths:
1. *Novel and well-motivated problem setting*: The paper focuses on unsupervised FGL, a relatively under-explored yet practically important setting where label scarcity and data heterogeneity present serious challenges.
2. *Well-organized methodology*: The proposed method is intuitive and reasonable, including both RSA and AGPL from two perspectives.
3. *Clear writing and presentation*: Despite the technical depth, the paper is generally well-written with clear motivation, formulation, and visual illustrations (e.g., Figure 1–3).

Weaknesses:
1. *Limited theoretical support*: While the paper proposes technically rich modules, it lacks formal analysis (e.g., convergence, generalization bounds, or even complexity analysis in the main text), which would have strengthened the methodological contribution.
2. *Computational overhead and scalability unclear*: The main paper lacks discussion on training cost, communication overhead, or scalability in practical federated environments.
3. *Out-dated baselines*: Only one baseline, FedU$^2$, was proposed after 2023. More advanced baselines should be included to demonstrate the advantage of proposed method. Moreover, some remarkable related works, such as [1-3], are missing, which need to be added and discussed.

[1] Federated Contrastive Learning of Graph-Level Representations.

[2] Federated Self-Explaining GNNs with Anti-shortcut Augmentations.

[3] Subgraph Federated Unlearning.

---

> ### Author Rebuttal · Authors · 2025-07-31
>
> **R1 to W1**: Thank you for this constructive advice.
> For a comprehensive understanding of the proposed FedPAM, we provide the analysis for the **generalization bounds** and **complexity analysis**. Given a FL system, its generalization bound for the global model is defined as **Lemma 1**.
>
> >**Lemma 1**. Given a FL system with global distribution ${\mathcal{D}}$ and local distribution ${\mathcal{D}}_{k}$, the generalization error for any hypothesis with the probability at least $1-\delta$ ($0<\delta\leq 1$) is
>
> $$
> \mathcal{R}\_{\mathcal{D}}(h) \leq \frac{1}{K} \sum\_{k \in[K]} \hat{\mathcal{R}}\_{\tilde{\mathcal{D}}\_k}\left(h\_k\right)+\frac{1}{K} \sum_{k \in[K]}\left(d\_{\mathcal{H} \Delta \mathcal{H}}\left(\tilde{\mathcal{D}}\_k, \tilde{\mathcal{D}}\right)+\lambda\_k\right)
> +\sqrt{\frac{4}{m}\left(d \log \frac{2 e m}{d}+\log \frac{4 K}{\delta}\right)}.
> $$
>
> For the $\mathcal{H}$-divergence $d_{\mathcal{H}\Delta\mathcal{H}}(\tilde{\mathcal{D}}_{k},\tilde{\mathcal{D}})$, we further have
>
> $$
> \begin{aligned}
> 	d\_{\mathcal{H} \Delta \mathcal{H}}\left(\tilde{\mathcal{D}}\_k, \tilde{\mathcal{D}}\right) & \leq d\_{\mathcal{H} \Delta \mathcal{H}}\left(\tilde{\mathcal{D}}\_k, \tilde{\mathcal{D}}\_l\right)+d\_{\mathcal{H} \Delta \mathcal{H}}\left(\tilde{\mathcal{D}}\_l, \tilde{\mathcal{D}}\right) (K-1) d\_{\mathcal{H} \Delta \mathcal{H}}\left(\tilde{\mathcal{D}}\_k, \tilde{\mathcal{D}}\right) & \leq \sum\_{l \neq k}^K\left[d\_{\mathcal{H} \Delta \mathcal{H}}\left(\tilde{\mathcal{D}}\_k, \tilde{\mathcal{D}}\_l\right)+d\_{\mathcal{H} \Delta \mathcal{H}}\left(\tilde{\mathcal{D}}\_l, \tilde{\mathcal{D}}\right)\right] \\
> \end{aligned}
> $$
> Assume $\forall l \in [K]$, $d\_{\mathcal{H}\Delta\mathcal{H}}(\tilde{\mathcal{D}}\_{l},\tilde{\mathcal{D}})\leq \epsilon\_{0}$, it can obtain
> $$
> (K-1) d\_{\mathcal{H} \Delta \mathcal{H}}\left(\tilde{\mathcal{D}}\_k, \tilde{\mathcal{D}}\right)
> \leq \sum\_{l \neq k}^K\left[d\_{\mathcal{H} \Delta \mathcal{H}}\left(\tilde{\mathcal{D}}\_k, \tilde{\mathcal{D}}\_l\right)+\epsilon\_0\right]
> $$
>
> $$
> \frac{1}{K} \sum\_{k \in[K]} d\_{\mathcal{H} \Delta \mathcal{H}}\left(\tilde{\mathcal{D}}\_k, \tilde{\mathcal{D}}\right)
> \leq \frac{1}{K(K-1)} \sum\_{k \in[K]} \sum\_{l \neq k}^K\left[d\_{\mathcal{H} \Delta \mathcal{H}}\left(\tilde{\mathcal{D}}\_k, \tilde{\mathcal{D}}\_l\right)+\epsilon\_0\right]\\
> $$
>
> $$
> \leq \frac{1}{K(K-1)}\sum\_{k \in[K]} \sum\_{l \neq k}^Kd\_{\mathcal{H} \Delta \mathcal{H}}\left(\tilde{\mathcal{D}}\_k, \tilde{\mathcal{D}}\_l\right) +\underbrace{\frac{1}{K(K-1)} \sum\_{k \in[K]} \sum\_{l \neq k}^K \epsilon\_0}\_\epsilon.
> $$
>
> Then, we have following **Theorem**
>
> >**Theorem 1**. Given a FGL system with global distribution ${\mathcal{D}}$ and local distribution ${\mathcal{D}}\_{k}$, the generalization error for any hypothesis with the probability at least $1-\delta$ ($0<\delta\leq 1$) is
>  $$
> \mathcal{R}\_{{\mathcal{D}}}(h) \leq \frac{1}{K} \sum\_{k \in[K]} \hat{\mathcal{R}}\_{\tilde{\mathcal{D}}\_k}\left(h\_k\right)
> +\frac{1}{K(K-1)} \sum\_{k \in[K]} \sum\_{l \neq k}^K d\_{\mathcal{H} \Delta \mathcal{H}}\left(\tilde{\mathcal{D}}\_k, \tilde{\mathcal{D}}\_l\right)+\epsilon+\frac{1}{K} \sum\_{k \in[K]} \lambda\_k+\sqrt{\frac{4}{m}\left(d \log \frac{2 e m}{d}+\log \frac{4 K}{\delta}\right)}.
> $$
>
> According to the GNN's generalization analysis in [1],
> we can obtain the upper bound of $d\_{\mathcal{H}\Delta\mathcal{H}}(\tilde{\mathcal{D}}\_{k},\tilde{\mathcal{D}}\_{l})$:
> $$
> d\_{\mathcal{H} \Delta \mathcal{H}}\left(\tilde{\mathcal{D}}\_k, \tilde{\mathcal{D}}\_l\right) \\
> $$
>
> $$
> \leq 1+\frac{B\_{rank}}{m} \sup \_f\left\|f\left(G\_k\right)-f\left(G\_l\right)\right\|\_F \\
> $$
>
> $$
> \leq 1+\sup \_f B\_W^2 B\_X\left(\frac{D\_{\min }+m}{D\_{\min }^2 m}\right)\left\|\mathbf{A}\_k-\mathbf{A}\_l\right\|\_F+\frac{B\_W^2}{m}\left\|\mathbf{X}\_k-\mathbf{X}\_l\right\|\_F
> $$
> where $B\_{rank}$, $B_{X}$, $D_{min}$ denote relevant constants,
> $\left\|\mathbf{A}_k-\mathbf{A}_l\right\|_F$ and $\left\|\mathbf{X}_k-\mathbf{X}_l\right\|_F$ measure the divergences between different topologies and features, respectively. Then, we can see that when the topological and feature differences of different client subgraphs are reduced, the generalization ability of the global model will be enhanced. The proposed FedPAM uses FGW-OT to achieve the optimal transport between the global anchor graph and each subgraph, which objectively reduces the differences between different subgraphs. Therefore, the generalization ability of the proposed FedPAM will be improved.
>
> [1] Towards understanding generalization of graph neural networks.
>
> **Analysis for Computational Complexity:** The computation complexity mainly originates from two aspects: the FGW-OT for aligning representation spaces and the low-rank tensor optimization for adaptively learning global model parameters.
> For the FGW-OT, the computation of $G$ (an intermediate variable) takes $\mathcal{O}(N_{E}M+M^{2}N)$, where $N_{E}$ denotes the number of edges, $M$ denotes the number of anchors, and $N$ denotes the number of nodes.
> Notably, the adjacency matrices are often sparse, so the computational complexity can be significantly decreased.
> Likewise, the OT based on Sinkhorn algorithm takes $\mathcal{O}(MN)$.
> Then, the FGW-OT takes $\mathcal{O}(N_{E}M+M^{2}N+MN)$.
> For the low-rank tensor optimization, the update of $\boldsymbol{\mathcal{G}}$ with $D_{l,1}\times D_{l,2}\times K$ occupies the dominant computational complexity.
> First, the FFT and the inverse FFT require $\mathcal{O}(D_{l,1}^{2}D_{l,2}\log(K))$, where $D_{l,1}$ and $D_{l,2}$ are the dimensions for the $l$-th layer's parameters, $K$ is the number of clients, and $D_{l,1}$ is the higher dimension.
> Second, the T-SVD costs $\mathcal{O}(D_{l,1}D_{l,2}^{2}K)$. Then, the low-rank tensor optimization takes $\mathcal{O}(D_{l,1}^{2}D_{l,2}\log(K)+D_{l,1}D_{l,2}^{2}K)$.
> Overall, the computational complexity for the proposed FedPAM is $\mathcal{O}(N_{E}M+M^{2}N+MN+D_{l,1}^{2}D_{l,2}\log(K)+D_{l,1}D_{l,2}^{2}K)$.
>
> **R2 to W2:** Thanks for this good comment. We have explained the computational complexity in \textbf{R1}.
> For the proposed FedPAM, its communication overhead is $2\times(K|\mathbf{W}\_{k}|+KML)$, where $|\mathbf{W}\_{k}|$ denotes the model parameter volume, $L$ denotes the dimension of anchors.
> Compared to other FGL methods, FedPAM additionally transmit a set of anchors, which are essentially multiple feature vectors and do not take up much communication traffic.
> For the scalability in practical federated environments, we test FedPAM with large-scale clients, e.g. 50 and 100 clients. The experimental results are reported in Tables 1 and 2. We can see that FedPAM still achieves superior performance.
>
> *Table 1: Performance comparison with 50 clients, where the SimCLR is employed in local training.*
> | Method   | PubMed (ACC) | PubMed (Fscore) | OA (ACC) | OA (Fscore) |
> |----------|------------|---------------|--------|-----------|
> | FedAvg   | 73.57      | 65.15         | 43.59  | 31.57     |
> | FedProx  | 70.42      | 59.40         | 40.83  | 28.98     |
> | MOON     | 70.44      | 59.45         | 42.78  | 30.44     |
> | FedX     | 70.43      | 59.43         | 47.81  | 34.83     |
> | FedU$^2$ | 72.65      | 63.60         | 44.07  | 30.62     |
> | **FedPAM** | **74.29** | **67.31**     | **49.85** | **38.34** |
>
> *Table 2: Performance comparison with 100 clients, where the SimCLR is employed in local training.*
> | Method   | PubMed (ACC) | PubMed (Fscore) | OA (ACC) | OA (Fscore) |
> |----------|------------|---------------|--------|-----------|
> | FedAvg   | 68.21      | 56.76         | 37.90  | 26.01     |
> | FedProx  | 68.23      | 56.78         | 36.18  | 24.13     |
> | MOON     | 68.08      | 56.62         | 38.84  | 26.95     |
> | FedX     | 69.27      | 58.25         | 47.04  | 33.18     |
> | FedU$^2$ | 71.82      | 62.66         | 38.08  | 26.23     |
> | **FedPAM** | **72.36** | **65.22**     | **48.58** | **37.32** |
>
> **R3 to W3:** Thank you for this valuable suggestion. We added a new unsupervised FL algorithm, namely FLPD [1]. The experimental results are shown in Table 3. It can be seen that the proposed FedPAM achieves more superior performance.
> The differences between FedPAM and the three works [2][3][4] are from two aspects.
> First, we propose a new framework for unsupervised federated subgraph learning. To the best of our knowledge, we are the first to study this task. Second, the two proposed modules, RSA and AGPL, are novel and can well address the challenges of unsupervised federated graphs.
>
> *Table3: Performance comparison with a new unsupervised FL method FLPD.*
> | SSL       | Method  | Cora ACC | Cora Fscore | CiteSeer ACC | CiteSeer Fscore |
> |-----------|---------|----------|-------------|--------------|-----------------|
> | **Simsiam** | FLPD    | 57.27    | 45.43       | 40.63        | 26.41           |
> |           | FedPAM  | **61.40**| **52.85**   | **49.53**    | **38.73**       |
> | **SimCLR**  | FLPD    | 56.95    | 43.22       | 38.91        | 22.87           |
> |           | FedPAM  | **58.62**| **45.21**   | **43.90**    | **31.61**       |
> | **BYOL**    | FLPD    | 65.37    | 54.01       | 44.82        | 32.34           |
> |           | FedPAM  | **66.01**| **55.23**   | **48.81**    | **38.69**       |
>
>
> [1] Prototype similarity distillation for communication-efficient federated unsupervised representation learning
>
> [2] Federated Contrastive Learning of Graph-Level Representations
>
> [3] Federated Self-Explaining GNNs with Anti-shortcut Augmentations
>
> [4] Subgraph Federated Unlearning

---

> > ### Comment · Reviewer_cZbY · 2025-08-04
> >
> > Thank you for your response. After careful consideration, I decide to remain my score, i.e., boardline acc.

---

> > > ### Author Response · Authors · 2025-08-04
> > > **Response**
> > >
> > > Thank you again for provding many valuable comments and reviewing our responses！

---

### Comment · Area_Chair_5rrS · 2025-08-04

Dear Reviewers,

Please review the authors' rebuttal and indicate whether your concerns have been adequately addressed or not. Thank you for your efforts.

Best regards,

AC

---

### Decision · Program_Chairs · 2025-09-17

**Decision:**

Accept (poster)

**Comment:**

The work proposed a federated graph learning method via incorporating FGW and tensor decomposition. The experiments showed the effectiveness in comparison to a few baselines. I summarize the strengths and weaknesses below.

**Strengths**
1. The idea of using FGW and tensor decomposition for graph federated learning is novel.
2. The paper is easy to follow and well-organized.
3. The baselines and datasets in the experiments are sufficient. The results demonstrated the superiority of the proposed method over a few baselines.

**Weakness**
* The theoretical analysis is limited, though the authors tried to analyze the generalization bounds during the rebuttal.

During the rebuttal, the authors addressed most concerns of the reviewers, such as the computational complexity, running time, and more baselines. During the discussion, most reviewers supported accepting the paper.

In addition to the reviews given by the reviewers, I have the following comment. During the rebuttal, the authors tried to use the theoretical results of domain adaptation to derive a generalization bound for the proposed model. However, the theory usually requires an independence assumption, which does not hold for the nodes on a graph, because they are connected. In addition, using $|A_k-A_l|$ to measure the topological divergence makes no sense, since they may have different numbers of nodes and are permutation invariant. So, if the paper is accepted, the authors should be careful about the added theoretical results from the rebuttal.

Given the novel idea, the impressive numerical performance, the recognition of the reviewers, and the fact that the theoretical result is not compulsory for the current work, I recommend accepting the paper.